# Phase Field Models for Thermal Fracturing and Their Variational Structures

**DOI:** 10.3390/ma15072571

**Published:** 2022-03-31

**Authors:** Sayahdin Alfat, Masato Kimura, Alifian Mahardhika Maulana

**Affiliations:** 1Physics Education Department, Halu Oleo University, Kendari 93232, Southeast Sulawesi, Indonesia; 2Division of Mathematical and Physical Sciences, Graduate School on Natural Science and Technology, Kanazawa University, Kakuma, Kanazawa 920-1192, Japan; fiansmamda@gmail.com; 3Faculty of Mathematics and Physics, Kanazawa University, Kakuma, Kanazawa 920-1192, Japan; mkimura@se.kanazawa-u.ac.jp

**Keywords:** thermoelasticity, crack propagation, crack path, phase field model, variational structure, energy equality, adaptive finite element method

## Abstract

It is often observed that thermal stress enhances crack propagation in materials, and, conversely, crack propagation can contribute to temperature shifts in materials. In this study, we first consider the thermoelasticity model proposed by M. A. Biot and study its energy dissipation property. The Biot thermoelasticity model takes into account the following effects. Thermal expansion and contraction are caused by temperature changes, and, conversely, temperatures decrease in expanding areas but increase in contracting areas. In addition, we examine its thermomechanical properties through several numerical examples and observe that the stress near a singular point is enhanced by the thermoelastic effect. In the second part, we propose two crack propagation models under thermal stress by coupling a phase field model for crack propagation and the Biot thermoelasticity model and show their variational structures. In our numerical experiments, we investigate how thermal coupling affects the crack speed and shape. In particular, we observe that the lowest temperature appears near the crack tip, and the crack propagation is accelerated by the enhanced thermal stress.

## 1. Introduction

Cracking is a phenomenon that occurs everywhere in our lives, but, if it is allowed to continue, it can cause fatal damage. A crack in a material occurs when the material experiences a continuous overload. However, several other factors, such as thermal expansion and contraction due to temperature changes [1,2,3], fluid pressure (e.g., in hydraulic fracturing) [4], the diffusion of hydrogen (or hydrogen embrittlement) [5,6], chemical reactions [7], and humidity [2], cause cracks in materials. In particular, among these phenomena, cracks due to thermal expansion are interesting to study from the viewpoint of the energy balance between elastic, thermal, and surface energies.

M. A. Biot proposed a theoretical framework for coupled thermoelasticity based on the principle of minimum entropy production [8]. Biot’s model is now widely known as the traditional coupled thermoelasticity model, and it has been extended to dynamical theory [9] and to various other situations [10,11,12,13,14,15]. As shown in Section 2.2, it satisfies an energy balance equality between the elastic and thermal energies.

In fracture mechanics, especially in the modeling and simulation of crack propagation, a phase field approach has been recently recognized as a powerful tool. The phase field model (PFM) for fractures was first proposed by Bourdin et al. [16] and Karma et al. [17]. Then, based on the framework of variational fracture theory [18,19], the techniques and applications of PFM have been extensively developed, for example [20,21,22,23,24,25]. We refer to [26] for further information on the development of PFM for fracture mechanics. PFM for fracture mechanics is derived as a gradient flow of the total energy, which consists of the elastic energy and the surface energy and is known to be consistent with the classical Griffith theory [16,26]. It allows us to handle the complex geometry of multiple, kinked, or branching cracks in both 2D and 3D without a crack path search. Comparisons with the experimental results are investigated in [27].

The aim of our paper is propose an energy-consistent PFM for thermal fracturing by the coupling Biot thermoelasticity model. Naturally, three kinds of energy, i.e., elastic, thermal, and surface energies, appear in our stage, and the exchange and dissipation of those energies are the main interests of our research. An illustration is shown in Figure 1. There are several previous works that address thermal fracturing using PFM [26,28,29,30,31]. In particular, Miehe et al. [23] developed a theory of thermomechanical fracture with a diffusive crack model including various nonlinear effects and demonstrated several interesting numerical examples. However, the strain’s influence on the heat transfer as the Biot model is not involved in those previous works. It means that the previous model can not capture thermal distribution during crack propagation. To the best of our knowledge for the condition, a peridynamics model that employs the coupled thermoelastic equation was proposed by Gao and Oterkus [32]. In this study, for each PDE model, we consider an initial-value and boundary-value problem in a bounded domain 
$R^{d}$

(d=2or3)
 and derive its energy equality which represents the energy dissipation property.

The organization of this paper is as follows: in Section 2, we introduce the linear thermoelasticity model by M.A. Biot and derive its variational principle and energy dissipation property. In addition, we numerically investigate the effect of the thermal coupling term on the elastic and thermoelastic energies in an expanding region.

Section 3 is devoted to PFMs for crack propagation under thermal stress. In Section 3.1, we give a brief review of the irreversible fracturing phase field model (F-PFM) and its energy equality, which guarantees the energy dissipation property (Theorem 2) and follows the works [22,26]. In Section 3.2 and Section 3.3, we propose two types of thermal fracturing phase field models (TF-PFMs). The first model, TF-PFM1, is a straightforward coupling of F-PFM and the Biot thermoelasticity model. Based on the variational principle of the Biot model (Proposition 2), we show a partial energy equality for a fixed temperature (Theorem 3). However, it does not satisfy the energy equality for the total energy, which consists of the elastic, thermal, and surface energies.

The second model, TF-PFM2, presented in Section 3.3 is another natural coupling of F-PFM and the Biot thermoelasticity model based on the energy equality of the Biot model (Theorem 1). We prove an energy equality for TF-PFM2 in Theorem 4. Since we consider several models (Biot’s model, F-PFM, and TF-PFMs) and their energy qualities, for the readers’ convenience, we list the energies and energy equalities for each model in Table 1 and Table 2.

In Section 4, we show some numerical comparisons between two TF-PFMs using non-dimensionalized equations. We investigate the effects of the thermal coupling in TF-PFM1 and TF-PFM2 on the crack speed and the crack path by changing a dimensionless coupling parameter 
δ
. As noted, although the temperature influences material properties micro-structurally [33], it is not considered in the present study for simplicity. Generally, the effect of micro-structure of material gives the typical crack path, such as: curving, branching, kinking, etc. A clear study of this is addressed by [34]. The last section shows some conclusions and comments on further topics.

To easily understand the relevant notation and symbols in this paper, we introduce them in this section. Let 
Ω
 be a bounded domain in 
$R^{d}$
 (
d=2
 or 3). The position in 
$R^{d}$
 is denoted by 
$x={\left(x_{1}.\ldots ,x_{d}\right)}^{T}\in R^{d}$
, where 
${\;}^{T}$
 denotes the transposition of a vector or matrix. Let ∇, 
div
, and 
Δ
 be the gradient, divergence, and Laplacian operators with respect to *x*, respectively. For simplicity, we write 
$\dot{u}$
, 
$\dot{\Theta }$
, and 
$\dot{z}$
 as the partial derivatives of *u*, 
Θ
 and *z* with respect to *t*, respectively. For simplicity, we often denote 
u(t):=u(·,t)
, etc. The space of the real-valued (symmetric) 
d×d
 matrix is denoted by 
$R^{d\times d}$
 (
$R_{sym}^{d\times d}$
). The inner product of square matrices 
$A,B\in R^{d\times d}$
 is denoted by 
$A:B:=\sum_{i,j=1}^{d}A_{ij}B_{ij}$
. Using 
$L^{2}\left(\Omega \right)$
, we refer to the Lebesgue space on 
Ω
, while 
$H^{1}\left(\Omega ,R^{d}\right)$
 and 
$H^{\frac{1}{2}}\left(\Gamma_{D}^{u},R^{d}\right)$
 represent the Sobolev space on 
Ω
 and its trace space on the boundary 
$\Gamma_{D}^{u}$
, respectively. For more details on Sobolev spaces, we refer to the review in [35]. In addition, we summarize the physical properties used in this paper in Table 3.

## 2. Thermoelasticity Model

### 2.1. Formulation of the Problem

M.A. Biot [8] proposed the following mathematical model for coupled thermoelasticity:
(1)
$$\begin{array}{cc} \mathrm{div}\sigma \left[u\right]=\beta \nabla \Theta & \mathrm{in}\Omega \times [0,T], \end{array}$$

(2)
$$\begin{array}{cc} \chi \frac{\partial }{\partial t}\Theta =\kappa_{0}\Delta \Theta -\Theta_{0}\beta \frac{\partial }{\partial t}\left(\mathrm{div}u\right) & \mathrm{in}\Omega \times (0,T], \end{array}$$

where 
Ω
 is a bounded domain in 
$R^{d}$
 (
d=2
 or 3). We suppose that 
Ω
 is an isotropic elastic body and consider the thermoelastic coupling between the mechanical deformation and the thermal expansion in 
Ω
. The constant 
β
 is defined by 
$\beta :=a_{L}\left(d\lambda +2\mu \right)$
 with 
$a_{L}>0$
 as the coefficient of linear thermal expansion and 
μ(>0)
; 
$\lambda (>-\frac{2\mu }{d})$
 are Lamé’s constants.

The unknown functions in (1) and (2) are the displacement 
$u\left(x,t\right)=(u_{1}\left(x,t\right),$

$\ldots ,u_{d}{\left(x,t\right))}^{T}\in R^{d}$
 and the temperature 
Θ(x,t)∈R
. In addition, the constant 
$\Theta_{0}>0$
 is a fixed reference temperature. Similarly, strain 
e[u]
 and stress tensors 
σ[u]
 are defined as

$$\begin{array}{cc} e\left[u\right]:=\frac{1}{2}\left(\nabla u^{T}+{\left(\nabla u^{T}\right)}^{T}\right)\in R_{sym}^{d\times d}, & \left(3a\right)\\ \sigma \left[u\right]:=Ce\left[u\right]=\lambda \left(\mathrm{div}u\right)I+2\mu e\left[u\right]\in R_{sym}^{d\times d}, & \left(3b\right) \end{array}$$

where 
$C:=\left(c_{ijkl}\right),c_{ijkl}=\lambda \delta_{ij}\delta_{kl}+\mu \left(\delta_{ik}\delta_{jl}+\delta_{il}\delta_{jk}\right)$
 is an isotropic elastic tensor and *I* is the identity matrix of size *d*. From (3b), (1) is also written in the form

$$\begin{array}{cc} & \mu \Delta u+(\lambda +\mu )\nabla (\mathrm{div}u)=\beta \nabla \Theta . \end{array}$$


The term 
β∇Θ
 in (1) and the term 
$\Theta_{0}\beta \frac{\partial }{\partial t}\left(\mathrm{div}u\right)$
 in (2) represent the body force due to thermal expansion and the heat source due to the volume change rate, respectively. We remark that, when 
$a_{L}=0$
, (1) and (2) are decoupled.

It is convenient to introduce the following strain and stress tensors, including the thermal effect:

$$\begin{array}{cc} e^{*}\left[u,\Theta \right]:=e\left[u\right]-a_{L}\left(\Theta \left(x,t\right)-\Theta_{0}\right)I\in R_{sym}^{d\times d}, & \left(4a\right)\\ \sigma^{*}\left[u,\Theta \right]:=Ce^{*}\left[u,\Theta \right]=\sigma \left[u\right]-\beta \left(\Theta \left(x,t\right)-\Theta_{0}\right)I\in R_{sym}^{d\times d}. & \left(4b\right) \end{array}$$


Using the thermal stress tensor 
$\sigma^{*}\left[u,\Theta \right]$
, (1) can be written in the following form:
$$\begin{array}{cc} & -\mathrm{div}\sigma^{*}\left[u,\Theta \right]=0. \end{array}$$


This means that the force 
$\sigma^{*}\left[u,\Theta \right]$
 is in equilibrium in 
Ω
. In the preceding, Equations (1) and (2) represent the force balance and the thermal diffusion in 
Ω
, respectively.

The system in (1) and (2) is complemented by the following boundary and initial conditions:

$$\left\{ \begin{array}{ccc} u=u_{D}\left(x,t\right) & \mathrm{on}\Gamma_{D}^{u}\times \left[0,T\right], & \left(5a\right)\\ \sigma^{*}\left[u,\Theta \right]n=0 & \mathrm{on}\Gamma_{N}^{u}\times \left[0,T\right], & \left(5b\right)\\ \Theta =\Theta_{D}\left(x,t\right) & \mathrm{on}\Gamma_{D}^{\Theta }\times \left[0,T\right], & \left(5c\right)\\ \frac{\partial \Theta }{\partial n}=0 & \mathrm{on}\Gamma_{N}^{\Theta }\times \left[0,T\right], & \left(5d\right)\\ \Theta \left(x,0\right)=\Theta_{*}\left(x\right) & \mathrm{in}\Omega , & \left(5e\right) \end{array}\right.$$

where *n* is the outward unit normal vector along the boundary, 
$\Gamma =\Gamma_{D}^{u}\cup \Gamma_{N}^{u}$

$\left(\Gamma =\Gamma_{D}^{\Theta }\cup \Gamma_{N}^{\Theta }\right)$
 with 
$\Gamma_{D}^{u}\cap \Gamma_{N}^{u}=\emptyset \left(\Gamma_{D}^{\Theta }\cap \Gamma_{N}^{\Theta }=\emptyset \right)$
. The boundaries 
$\Gamma_{D}^{u}$
 and 
$\Gamma_{N}^{u}$
 (
$\Gamma_{D}^{\Theta }$
 and 
$\Gamma_{N}^{\Theta }$
) are the Dirichlet and Neumann boundaries for *u* (for 
Θ
), respectively. We suppose that the 
(d−1)
-dimensional volume of 
$\Gamma_{D}^{u}$
 is positive for the solvability of *u*.

**Remark** **1.**Instead of boundary conditions (5a) and (5b), we can also consider the following mixed-type condition. When 
d=2
, on a part of the boundary (which we denote by 
$\Gamma_{DN}^{u}$
), 
$u={\left(u_{1},u_{2}\right)}^{T}$
 and

$$\begin{array}{c} \left\{\begin{array}{cc} u_{1}=u_{D1} & \mathrm{on}\Gamma_{DN}^{u},\\ \left(\sigma^{*}\left[u,\Theta \right]n\right)\cdot e_{2}=0 & \mathrm{on}\Gamma_{DN}^{u}, \end{array}\right. \end{array}$$

or

$$\begin{array}{c} \left\{\begin{array}{cc} u_{2}=u_{D2} & \mathrm{on}\Gamma_{DN}^{u},\\ \left(\sigma^{*}\left[u,\Theta \right]n\right)\cdot e_{1}=0 & \mathrm{on}\Gamma_{DN}^{u}, \end{array}\right. \end{array}$$

where 
$u_{Di}:=\Gamma_{DN}^{u}\mapsto R$
 is a given horizontal or vertical displacement and 
$e_{1}={\left(1,0\right)}^{T}$
, 
$e_{2}={\left(0,1\right)}^{T}$
. These types of mixed boundary conditions are considered in Section 2.3.3 and Section 4.4.1. Even for these mixed-type boundary conditions, we can easily extend the following arguments on weak solutions, variational principles, and energy equalities.

### 2.2. Variational Principle and Energy Equality

This section aims to show a variational principle and provide an energy equality that implies the energy dissipation property for the system (1) and (2). In linear elasticity theory, a weak form of the boundary value problem for 
$u_{D}\in H^{\frac{1}{2}}\left(\Gamma_{D}^{u};R^{d}\right)$
 is

(6)
$$\begin{array}{c} \left\{\begin{array}{cc} -\mathrm{div}\sigma [u]=0 & \mathrm{in}\Omega ,\\ u=u_{D} & \mathrm{on}\Gamma_{D}^{u},\\ \sigma [u]n=0 & \mathrm{on}\Gamma_{N}^{u}, \end{array}\right. \end{array}$$

which is given by

$$\begin{array}{cc} & u\in V^{u}\left(u_{D}\right),\int_{\Omega }\sigma \left[u\right]:e\left[v\right]dx=0\;\mathrm{for}\mathrm{all}v\in V^{u}\left(0\right), \end{array}$$

where

(7)
$$\begin{array}{cc} & V^{u}\left(u_{D}\right):=\left\{u\in H^{1}\left(\Omega ;R^{d}\right);u{|}_{\Gamma_{D}^{u}}=u_{D}\right\}. \end{array}$$


A weak solution uniquely exists and is given by

$$u=\underset{v\in V^{u}\left(u_{D}\right)}{\mathrm{argmin}}E_{el}\left(v\right),$$

where

(8)
$$\begin{array}{cc} & E_{el}\left(v\right):=\frac{1}{2}\int_{\Omega }\sigma \left[v\right]:e\left[v\right]dx\;(v\in H^{1}\left(\Omega ;R^{d})\right) \end{array}$$

is an elastic energy. This is known as a variational principle [37,38]. For a fixed 
Θ(x)
, a weak form for *u* of (1) and its variational principle are derived as follows:

**Proposition** **1.***For 
$u\in H^{2}\left(\Omega ;R^{d}\right)$
 and 
$\Theta \in H^{1}\left(\Omega \right)$
,*

$$\begin{array}{c} \left\{\begin{array}{cc} -\mathrm{div}\sigma^{*}\left[u,\Theta \right]=0 & \mathrm{in}\Omega ,\\ u=u_{D} & \mathrm{on}\Gamma_{D}^{u},\\ \sigma^{*}\left[u,\Theta \right]n=0 & \mathrm{on}\Gamma_{N}^{u}, \end{array}\right. \end{array}$$

*is equivalent to the following weak form:*

(9)
$$\begin{array}{c} \left\{\begin{array}{cc} \int_{\Omega }\sigma^{*}\left[u,\Theta \right]:e\left[v\right]dx=0 & \mathrm{for}\mathrm{all}\;v\in V^{u}\left(0\right),\\ u\in V^{u}\left(u_{D}\right). \end{array}\right. \end{array}$$


**Proof.** For 
$v\in V^{u}\left(0\right)$
, we have

$$\begin{array}{ccc} \int_{\Omega }\left(-\mathrm{div}\sigma^{*}\left[u,\Theta \right]\right)\cdot v\;dx & = & \int_{\Omega }\sigma^{*}\left[u,\Theta \right]:e\left[v\right]dx-\int_{\Gamma_{N}^{u}}\left(\sigma^{*}\left[u,\Theta \right]n\right)\cdot v\;ds. \end{array}$$
The equivalency immediately follows from this equation: □

**Proposition** **2**(Variational principle)**.**
*For a given 
$\Theta \in L^{2}\left(\Omega \right)$
, 
$u_{D}\in H^{\frac{1}{2}}\left(\Gamma_{D}^{u};R^{d}\right)$
, there exists a unique weak solution 
$u\in H^{1}\left(\Omega ;R^{d}\right)$
 that satisfies (9). Furthermore, the solution u is a unique minimizer of the variational problem:*

$$u=\underset{v\in V^{u}\left(u_{D}\right)}{\mathrm{argmin}}E_{el}^{*}\left(v,\Theta \right),$$

*where*

(10)
$$\begin{array}{cc} & E_{el}^{*}\left(v,\Theta \right)=\frac{1}{2}\int_{\Omega }\sigma^{*}\left[v,\Theta \right]:e^{*}\left[v,\Theta \right]dx. \end{array}$$

*We remark that 
$E_{el}^{*}\left(v,\Theta \right)$
 represents thermoelastic energy.*


**Proof.** The unique existence of a weak solution for *u* is shown by the Lax–Milgram theorem [38] since (9) is written as

$$\begin{array}{c} \left\{\begin{array}{c} \int_{\Omega }\sigma \left[u\right]:e\left[v\right]dx=\int_{\Omega }\beta \left(\Theta -\Theta_{0}\right)\mathrm{div}v\;dx,\\ u\in V^{u}\left(u_{D}\right)\;\left(\mathrm{for}\mathrm{all}\;v\in V^{u}\left(0\right)\right). \end{array}\right. \end{array}$$
The coercivity of the above weak form is known as Korn’s second inequality [38]:

$$\begin{array}{cc} & {}^{\exists }a_{0}>0\mathrm{such}\mathrm{that}\int_{\Omega }\sigma \left[v\right]:e\left[v\right]dx\geq a_{0}{∥v∥}_{H^{1}\left(\Omega ;R^{2}\right)}^{2},\mathrm{for}\mathrm{all}\;v\in V^{u}\left(0\right). \end{array}$$
For a weak solution *u* and any 
$v\in V^{u}\left(0\right)$
, using the equalities

$$\begin{array}{cc} & \sigma^{*}\left[u+v,\Theta \right]=\sigma^{*}\left[u,\Theta \right]+\sigma \left[v\right],\\ & e^{*}\left[u+v,\Theta \right]=e^{*}\left[u,\Theta \right]+e\left[v\right],\\ & \sigma^{*}\left[u,\Theta \right]:e\left[v\right]=e^{*}\left[u,\Theta \right]:\sigma \left[v\right], \end{array}$$

we have

$$\begin{array}{cc} E_{el}^{*}\left(u+v,\Theta \right)-E_{el}^{*}\left(u,\Theta \right) & =\frac{1}{2}\int_{\Omega }\sigma^{*}\left[u+v,\Theta \right]:e^{*}\left[u+v,\Theta \right]dx-\frac{1}{2}\int_{\Omega }\sigma^{*}\left[u,\Theta \right]:e^{*}\left[u,\Theta \right]dx\\ & =\int_{\Omega }\sigma^{*}\left[u,\Theta \right]:e\left[v\right]dx+\frac{1}{2}\int_{\Omega }\sigma \left[v\right]:e\left[v\right]dx\\ & =\frac{1}{2}\int_{\Omega }\sigma \left[v\right]:e\left[v\right]dx\geq 0. \end{array}$$
This shows that *u* is a minimizer of 
$E_{el}^{*}\left(u,\Theta \right)$
 among 
$V^{u}\left(u_{D}\right)$
.On the other hand, if *u* is a minimizer, the first variation of 
$E_{el}^{*}$
 vanishes at *u*; i.e., for all 
$v\in V^{u}\left(0\right)$
, we have

$$\begin{array}{cc} & 0=\frac{d}{ds}E_{el}^{*}\left(u+sv,\Theta \right){|}_{s=0}=\int_{\Omega }\sigma^{*}\left[u,\Theta \right]:e\left[v\right]dx. \end{array}$$
Hence, *u* is a weak solution. Summarizing the above, there exists a unique weak solution to (8), and *u* is a weak solution if and only if it is a minimizer of 
$E_{el}^{*}$
 among 
$V^{u}\left(u_{D}\right)$
. □

The next theorem represents a dissipation of the sum of the elastic and thermal energies during the thermomechanical process. We define thermal energy as

(11)
$$\begin{array}{c} E_{th}\left(\Theta \right):=\frac{\chi }{2\Theta_{0}}\int_{\Omega }{\left|\Theta \left(x\right)-\Theta_{0}\right|}^{2}dx. \end{array}$$


**Theorem** **1**(Energy equality for Biot’s model)**.**
*Let 
(u(x,t),Θ(x,t))
 be a sufficiently smooth solution to (1), (2) and (5). In addition, we suppose that 
$u_{D}$
 does not depend on t and 
$\Theta_{D}=\Theta_{0}$
. Then,*

(12)
$$\begin{array}{cc} \frac{d}{dt}\left(E_{el}\left(u\left(t\right)\right)+E_{th}\left(\Theta \left(t\right)\right)\right) & =-\frac{\kappa_{0}}{\Theta_{0}}\int_{\Omega }{\left|\nabla \Theta (t)\right|}^{2}dx\leq 0. \end{array}$$


**Proof.** Since

(13)
$$\begin{array}{cc} \frac{d}{dt}\left(\frac{1}{2}\sigma \left[u\right]:e\left[u\right]\right) & =\sigma \left[u\right]:e\left[\dot{u}\right]\\ & =\left(\sigma^{*}\left[u,\Theta \right]+\beta \left(\Theta -\Theta_{0}\right)I\right):e\left[\dot{u}\right]\\ & =\sigma^{*}\left[u,\Theta \right]:e\left[\dot{u}\right]+\beta \left(\Theta -\Theta_{0}\right)\mathrm{div}\dot{u} \end{array}$$

we obtain

(14)
$$\begin{array}{cc} \frac{d}{dt}E_{el}\left(u\left(t\right)\right) & =\frac{1}{2}\int_{\Omega }\frac{d}{dt}\left(\sigma [u]:e[u]\right)dx\\ & =\int_{\Omega }\sigma^{*}\left[u,\Theta \right]:e\left[\dot{u}\right]dx+\int_{\Omega }\beta \left(\Theta -\Theta_{0}\right)\left(\mathrm{div}\dot{u}\right)dx\\ & =\int_{\Omega }\beta \left(\Theta -\Theta_{0}\right)\left(\mathrm{div}\dot{u}\right)dx. \end{array}$$
Substituting (2) into (14) and using the boundary conditions (5c) and (5d) for 
Θ
, we obtain

$$\begin{array}{cc} \frac{d}{dt}E_{el}\left(u\left(t\right)\right) & =\int_{\Omega }\left(\frac{1}{\Theta_{0}}\left(\Theta -\Theta_{0}\right)\left\{\kappa_{0}\Delta \Theta -\chi \frac{\partial \Theta }{\partial t}\right\}\right)dx\\ & =\frac{\kappa_{0}}{\Theta_{0}}\int_{\Gamma }\left(\Theta -\Theta_{0}\right)\frac{\partial \Theta }{\partial n}ds-\frac{\kappa_{0}}{\Theta_{0}}\int_{\Omega }|\nabla \Theta {|}^{2}dx-\frac{d}{dt}\left(\frac{\chi }{2\Theta_{0}}\int_{\Omega }|\Theta -\Theta_{0}{|}^{2}dx\right)\\ & =-\frac{\kappa_{0}}{\Theta_{0}}\int_{\Omega }|\nabla \Theta {|}^{2}dx-\frac{d}{dt}E_{th}\left(\Theta \left(t\right)\right). \end{array}$$
This gives the energy equality for (5). □

As shown in Proposition 2 and Theorem 1, Biot’s thermoelasticity model is related to both energies 
$E_{el}\left(u\right)$
 and 
$E_{el}^{*}\left(u,\Theta \right)$
. We denote their energy densities as follows:
(15)
$$\begin{array}{cc} & W(u):=\sigma [u]:e[u], \end{array}$$

(16)
$$\begin{array}{cc} & W^{*}\left(u,\Theta \right):=\sigma^{*}\left[u,\Theta \right]:e^{*}\left[u,\Theta \right], \end{array}$$

where 
W(u)
 and 
$W^{*}\left(u,\Theta \right)$
 are the elastic and thermoelastic energy densities, respectively.

### 2.3. Numerical Experiment

#### 2.3.1. Non-Dimensional Setting

In the following numerical examples, we introduce a non-dimensional form of Biot’s model. We consider the following scaling for *x*, *t*, *u*, *C* (or 
λ
, 
μ
), and 
Θ
:
(17)
$$\begin{array}{c} \tilde{x}=\frac{x}{c_{x}},\tilde{t}=\frac{t}{c_{t}},\tilde{u}=\frac{u}{c_{u}},\tilde{C}=\frac{C}{c_{e}},\tilde{\Theta }=\frac{\Theta -\Theta_{0}}{c_{\Theta }},{\tilde{a}}_{L}=\frac{c_{x}c_{\Theta }}{c_{u}}a_{L},\tilde{\beta }=1, \end{array}$$

where 
$c_{x}$
, 
$c_{t}$
, 
$c_{u}$
, 
$c_{e}$
, and 
$c_{\Theta }>0$
 are the scaling parameters. Let 
$c_{x}$
 [
m
], 
$c_{e}$
 [
Pa
], and 
$c_{\Theta }$
 [
K
] be characteristic scales for the length of the domain, the size of the elastic tensor, and the temperature, respectively. The parameters 
$c_{t}$
 and 
$c_{u}$
 are defined as

(18)
$$\begin{array}{c} c_{t}:=\frac{c_{x}^{2}\chi }{\kappa_{0}}\left[s\right],c_{u}:=\frac{c_{\Theta }c_{x}\beta }{c_{e}}\left[m\right], \end{array}$$

where 
χ
 [
$\mathrm{Pa}\cdot K^{-1}$
], 
$\kappa_{0}$
 [
$\mathrm{Pa}\cdot m^{2}\cdot s^{-1}\cdot K^{-1}$
] and 
$\beta =a_{L}\left(d\lambda +2\mu \right)$
 [
$\mathrm{Pa}\cdot K^{-1}$
]. Then, (1) and (2) are written in the following non-dimensional form:

$$\left\{ \begin{array}{ccc} \tilde{\mathrm{div}}\tilde{\sigma }\left[\tilde{u}\right]=\tilde{\nabla }\tilde{\Theta } & \mathrm{in}\tilde{\Omega }\times \left[0,\tilde{T}\right], & \left(19a\right)\\ \frac{\partial }{\partial \tilde{t}}\tilde{\Theta }=\tilde{\Delta }\tilde{\Theta }-\delta \frac{\partial }{\partial \tilde{t}}\left(\tilde{\mathrm{div}}\tilde{u}\right) & \mathrm{in}\tilde{\Omega }\times \left(0,\tilde{T}\right]. & \left(19b\right) \end{array}\right.$$


The system (19) has only three parameters, 
$\tilde{\lambda }$
, 
$\tilde{\mu }$
, and 
δ
. The parameter 
δ
 is a non-dimensional thermoelastic coupling parameter defined by

$$\begin{array}{ccc} \delta & := & \frac{\Theta_{0}\beta^{2}}{c_{e}\chi }\left[-\right], \end{array}$$

and 
δ>0
. If we choose 
δ=0
, (19b) is decoupled from (19a), and the temperature field 
$\tilde{\Theta }$
 in (19a) is essentially a given function. In the following example, the case 
δ=0
 is referred to as the uncoupled case.

Under the above scaling, we denote the (thermo)elastic strain, stress tensors, and (thermo)elastic energy densities as follows:

$$\begin{array}{cc} \tilde{e}\left[\tilde{u}\right]:=\frac{1}{2}\left(\frac{\partial {\tilde{u}}_{i}}{\partial {\tilde{x}}_{j}}+\frac{\partial {\tilde{u}}_{j}}{\partial {\tilde{x}}_{i}}\right)=\frac{c_{x}}{c_{u}}e\left[u\right], & \left(20a\right)\\ \tilde{\sigma }\left[\tilde{u}\right]:=\tilde{C}\tilde{e}\left[\tilde{u}\right]=\frac{c_{x}}{c_{u}c_{e}}\sigma \left[u\right], & \left(20b\right)\\ \tilde{W}\left(\tilde{u}\right):=\tilde{\sigma }\left[\tilde{u}\right]:\tilde{e}\left[\tilde{u}\right]=\frac{c_{e}}{{\left(\beta c_{\Theta }\right)}^{2}}W\left[u\right], & \left(20c\right)\\ {\tilde{\sigma }}^{*}\left[\tilde{u},\tilde{\Theta }\right]:=\tilde{\sigma }\left[\tilde{u}\right]-\tilde{\Theta }I=\frac{1}{\beta c_{\Theta }}\sigma^{*}\left[u,\Theta \right], & \left(20d\right)\\ {\tilde{e}}^{*}\left[\tilde{u},\tilde{\Theta }\right]:=\tilde{e}\left[\tilde{u}\right]-{\tilde{a}}_{L}\tilde{\Theta }I=\frac{c_{x}}{c_{u}}\sigma^{*}\left[u,\Theta \right], & \left(20e\right)\\ {\tilde{W}}^{*}\left(\tilde{u},\tilde{\Theta }\right):={\tilde{\sigma }}^{*}\left[\tilde{u},\tilde{\Theta }\right]:{\tilde{e}}^{*}\left[\tilde{u},\tilde{\Theta }\right]=\frac{c_{e}}{{\left(\beta c_{\Theta }\right)}^{2}}W^{*}\left[u,\Theta \right]. & \left(20f\right) \end{array}$$


In the following section, we apply these non-dimensional forms and omit ∼ for simplicity.

#### 2.3.2. Numerical Setup and Time Discretization

In the following examples, we set Young’s modulus 
$E_{Y}=1$
, Poisson’s ratio 
$\nu_{P}=0.32$
, the coefficient of linear thermal expansion 
$a_{L}=0.475$
 and the thermoelasticity coupling parameter 
δ=0.0,0.1,0.5
 in the non-dimensional form of (19). We consider two numerical examples for (19), an L-shaped cantilever domain and a square domain with a crack (more precisely, a very sharp notch), as illustrated in Figure 2.

We apply the following implicit time discretization for (19):
(21)
$$\begin{array}{cc} & \left\{\begin{array}{cc} -\mathrm{div}\sigma^{*}\left[u^{k},\Theta^{k-1}\right]=0 & \mathrm{in}\Omega ,\\ \frac{\Theta^{k}-\Theta^{k-1}}{\Delta t}-\Delta \Theta^{k}+\delta \mathrm{div}\left(\frac{u^{k}-u^{k-1}}{\Delta t}\right)=0 & \mathrm{in}\Omega , \end{array}\right. \end{array}$$

where 
$u^{k}$
 and 
$\Theta^{k}$
 are approximations to *u* and 
Θ
 at 
t=kΔt(k=0,1,2,…)
. At each time step 
k=1,2,…
, we solve (21) with given boundary and initial conditions (5) using the finite element method. The details of the weak forms for (21) and their unique solvability are described in Appendix A.

In observation area 
A
 illustrated in Figure 2, we define the average of (thermo)elastic energy densities in 
A
 as follows:
$$\begin{array}{cc} W(A) & :=\frac{1}{\left|A\right|}\int_{A}W\left(u\right)dx,\\ W^{*}\left(A\right) & :=\frac{1}{\left|A\right|}\int_{A}W^{*}\left(u,\Theta \right)dx, \end{array}$$

and the differences between 
W(A)
 and 
$W^{*}\left(A\right)$
 for each 
δ>0
 and for 
δ=0
 are defined by

$$\begin{array}{c} \Delta W\left(A\right):=W\left(A\right){|}_{\delta }-W\left(A\right){|}_{\delta =0},\\ \Delta W^{*}\left(A\right):=W^{*}\left(A\right){|}_{\delta }-W^{*}\left(A\right){|}_{\delta =0}. \end{array}$$


In the following examples, we use the software FreeFEM [39] with P2 elements and unstructured meshes. For the time interval and time step, we use 
0≤t≤0.1
 and 
$\Delta t=1\times 10^{-4}$
, respectively.

#### 2.3.3. L-Shape Cantilever

Here, we consider the L-shaped cantilever whose left side is fixed, and the vertical displacement 
$u_{2}$
 is given on the right side, as illustrated in Figure 2 left. We denote the left and right boundaries by 
$\Gamma_{D}^{u}$
 and 
$\Gamma_{DN}^{u}$
, respectively, and define 
$\Gamma_{N}^{u}:=\Gamma \backslash \left(\Gamma_{D}^{u}\cup \Gamma_{DN}^{u}\right)$
. The boundary conditions for *u* are

$$\begin{array}{cc} & u=0,\mathrm{on}\Gamma_{D}^{u},\;\left\{\begin{array}{c} \sigma_{11}^{*}\left[u,\Theta \right]n=0,\\ u_{2}=-0.1t \end{array}\right.\mathrm{on}\Gamma_{DN}^{u},\;\sigma^{*}\left[u,\Theta \right]n=0\mathrm{on}\Gamma_{N}^{u}. \end{array}$$


For 
Θ
, we suppose 
$\frac{\partial \Theta }{\partial n}=0$
 on 
Γ
 and the initial temperature 
$\Theta_{*}=0$
. Although we adopt the above slightly modified boundary conditions in this example, the previous arguments are valid with small modifications, and we omit their details.

We apply the finite element method to (21). The total number of triangular meshes = 18,215 and the number of nodes (the vertices of the triangles) 
=9301
.

As shown in the lower part of Figure 3, we observe that the highest temperature is in the contracting area and the lowest is in the expanding area. Furthermore, there exists a contribution 
δ
 for each 
δ>0
 during the loading process. Although the disparity is small, the thermoelastic coupling parameter 
δ
 contributes to the variations in 
W(u)
 and 
$W^{*}\left(u\right)$
, as shown in Figure 4a,b. Here, a larger 
δ
 value implies larger 
W(A)
 and 
$W^{*}\left(A\right)$
 values (Figure 4d,e). In addition, we also observe that 
$W^{*}\left(A\right)$
 is larger than 
W(A)
 for each 
δ>0
 (Figure 4c).

In the L-shape cantilever case for each 
δ>0
, we conclude that the thermal coupling parameter enhances the singularity of (thermo)elastic energy in the expanding area. The (thermo)elastic energy plays a role in the driving force in the phase field model [23], which means that the parameter 
δ
 can accelerate crack growth in the expanding area.

#### 2.3.4. Cracked Domain

Here, we consider a cracked domain with vertical displacements on the top and bottom sides, and the other sides are free traction, as shown in Figure 2 right. The boundary conditions for *u* are

$$\begin{array}{c} \left\{\begin{array}{c} u_{1}=0,\\ u_{2}=\pm t \end{array}\right.\mathrm{on}\Gamma_{\pm D}^{u},\;\sigma^{*}\left[u,\Theta \right]n=0\mathrm{on}\Gamma_{N}^{u}, \end{array}$$

where 
$\Gamma_{+D}^{u}$
 and 
$\Gamma_{-D}^{u}$
 denote the top and bottom boundaries of 
Ω
, respectively, and 
$\Gamma_{N}^{u}:=\Gamma \backslash \left(\Gamma_{+D}^{u}\cup \Gamma_{-D}^{u}\right)$
. For 
Θ
, we suppose 
$\frac{\partial \Theta }{\partial n}=0$
 on 
$\Gamma_{N}^{\Theta }=\Gamma$
 and the initial temperature 
$\Theta_{*}=0$
.

We use the finite element method to solve (21). Therefore, the total number of triangular meshes and the number of nodes (the vertices of the triangles) are 11,176 and 5722, respectively.

From Figure 5 left, we conclude that the area that expands the most (i.e., 
divu
 is largest) appears near the crack tip. This can be compared with the analytical solution for the linear elasticity in a cracked domain in Appendix B. We also observe that the region with the lowest temperature appears to the right of the crack tip in Figure 5 right. From the temporal change in the temperature along the 
$x_{1}$
 axis plotted in Figure 6 right, we also observe that the lowest temperature region appears in 
$0.5<x_{1}<0.6$
 and that the temperature decreases over time. This is shown in Figure 6 left, where the value of 
divu
 is plotted along the 
$x_{1}$
 axis and 
divu
 is increasing over time; i.e., the heat source term 
$\mathrm{div}\dot{u}$
 in (2) is positive. Experimentally, the lowest temperature around the crack tips does not match with the studies of of Zehnder et al. [40], Rusinek et al. [41], and Wang et al. [42]. They record that the highest temperature occurs around the crack tips, which result from a plastic zone around the crack tips. In the present study, we do not consider the plastic zone. However, it would be more possible to use a thermo-viscous-elasticity condition. We will consider and study it using the thermo-viscous-elasticity equation in the future work [43].

Similar to Section 2.3.3, for each 
δ>0
, we obtain variations of 
W(A)
 and 
$W^{*}\left(A\right)$
 in subdomain 
A
 (Figure 7), where the subdomain 
A
 corresponds to the area that expands the most. From Figure 7, it is observed that 
$W^{*}\left(A\right)$
 is larger than 
W(A)
. This suggests that the thermoelastic energy density 
$W^{*}\left(u,\Theta \right)$
 has a higher value than the elastic energy density 
W(u)
. These observations are confirmed by the comparison of our thermal fracturing phase field models.

## 3. Crack Propagation under Thermal Stress

This section is devoted to the phase field models for thermal fracturing, which are the main purpose of this paper.

### 3.1. Fracturing Phase Field Model (F-PFM)

According to the works [22,26], we introduce fracturing PFM (we call it F-PFM) in this section. Let 
Ω
 be a bounded (uncracked) domain in 
$R^{d}$
 and 
$\Gamma :=\partial \Omega =\Gamma_{D}^{u}\cup \Gamma_{N}^{u}$
, similar to Section 2. In F-PFM, a crack in 
Ω
 at time *t* is described by a damage variable 
z(x,t)∈[0,1]
 for 
$x\in \bar{\Omega }$
 with space regularization. The cracked and uncracked regions are represented by 
z≈1
 and 
z≈0
, respectively, and 
z∈(0,1)
 indicates slight damage. A typical example of a straight crack in a square domain is illustrated in Figure 8.

The F-PFM is described as:

$$\left\{ \begin{array}{ccc} -\mathrm{div}\left({\left(1-z\right)}^{2}\sigma \left[u\right]\right)=0 & \mathrm{in}\Omega \times [0,T], & \left(22a\right)\\ \alpha \frac{\partial z}{\partial t}={\left(\epsilon \mathrm{div}\left(\gamma_{*}\nabla z\right)-\frac{\gamma_{*}}{\epsilon }z+\left(1-z\right)W\left(u\right)\right)}_{+} & \mathrm{in}\Omega \times [0,T], & \left(22b\right) \end{array}\right.$$

with the following boundary and initial conditions:

$$\left\{ \begin{array}{ccc} u=u_{D}\left(x,t\right) & \mathrm{on}\Gamma_{D}^{u}\times \left[0,T\right], & \left(23a\right)\\ \sigma [u]n=0 & \mathrm{on}\Gamma_{N}^{u}\times \left[0,T\right], & \left(23b\right)\\ \frac{\partial z}{\partial n}=0 & \mathrm{on}\Gamma \times [0,T], & \left(23c\right)\\ z\left(x,0\right)=z_{*}\left(x\right) & \mathrm{in}\Omega , & \left(23d\right) \end{array}\right.$$

where the displacement 
$u:\bar{\Omega }\times \left[0,T\right]\mapsto R^{d}$
 and the damage variable 
$z:\bar{\Omega }\times \left[0,T\right]\mapsto \left[0,1\right]$
 are unknowns. The parameters 
α>0
 and 
ϵ>0
 are small numbers related to regularization in time and space, respectively. The critical energy release rate is denoted by 
$\gamma_{*}$
 (which is often denoted by 
$G_{c}$
), and the elastic energy density is defined by 
W=W(u):=σ[u]:e[u]
. In (22b), the term *W* works as a driving force for *z*.

The symbol 
${\left(\;\right)}_{+}$
 on the right-hand side in (22b) denoted the positive part 
${\left(s\right)}_{+}:=\max\left(s,0\right)$
, and it represents the irreversible property of crack growth.

F-PFM is derived as a unidirectional gradient flow of the total energy 
$E_{el}\left(u,z\right)+E_{s}\left(z\right)$
, where

(24)
$$\begin{array}{cc} E_{el}\left(u,z\right) & :=\frac{1}{2}\int_{\Omega }{\left(1-z\right)}^{2}\sigma \left[u\right]:e\left[u\right]dx, \end{array}$$


(25)
$$\begin{array}{cc} E_{s}\left(z\right) & :=\frac{1}{2}\int_{\Omega }\gamma_{*}\left(\epsilon |\nabla z{|}^{2}+\frac{{\left|z\right|}^{2}}{\epsilon }\right)dx. \end{array}$$


More precisely, 
u(t)
 obeys the following variational principle:
(26)
$$u\left(t\right)=\underset{u\in V(u_{D}\left(t\right))}{\mathrm{argmin}}E_{el}\left(u,z\left(t\right)\right),$$

and (22b) becomes a gradient flow of the energy 
$\min_{u}E_{el}\left(u,z\right)+E_{s}\left(z\right)$
.

We remark that 
$E_{el}\left(u,z\right)$
 is a modified elastic energy, which corresponds to the elastic energy with a damaged Young’s modulus 
${\tilde{E}}_{Y}={\left(1-z\right)}^{2}E_{Y}$
. The energy 
$E_{s}\left(z\right)$
 is regularized surface energy, which approximates the crack area (
d=3
) or length (
d=2
) as 
ϵ↦0
. Please see [26] for more details. The following energy equality for F-PFM is shown in [26] ([22] for the antiplane setting).

**Theorem** **2**(Energy equality for F-PFM)**.**
*Let 
(u(x,t),z(x,t))
 be a sufficiently smooth solution to (22) and (23). If 
$u_{D}$
 is independent of t, then we have*

(27)
$$\begin{array}{c} \frac{d}{dt}\left(E_{el}\left(u\left(t\right),z\left(t\right)\right)+E_{s}\left(z\left(t\right)\right)\right)=-\alpha \int_{\Omega }{\left|\dot{z}\right|}^{2}dx\leq 0. \end{array}$$


**Proof.** Differentiating the total energy in *t* and applying integration by parts, we obtain

(28)
$$\begin{array}{cc} \frac{d}{dt} & \left(E_{el}\left(u\left(t\right),z\left(t\right)\right)+E_{s}\left(z\left(t\right)\right)\right)\\ & =\int_{\Omega }{\left(1-z\right)}^{2}\sigma \left[u\right]:e\left[\dot{u}\right]dx+\int_{\Omega }\left(\gamma_{*}\epsilon \nabla z\cdot \nabla \dot{z}+\left(\frac{\gamma_{*}}{\epsilon }z-\left(1-z\right)W\left(u\right)\right)\dot{z}\right)dx\\ & =\int_{\Gamma }{\left(1-z\right)}^{2}\underset{0}{\underbrace{\left(\sigma [u]n\right)}}\cdot \dot{u}\;ds-\int_{\Omega }\underset{0}{\underbrace{\mathrm{div}\left({\left(1-z\right)}^{2}\sigma \left[u\right]\right)}}\cdot \dot{u}\;dx+\int_{\Gamma }\gamma_{*}\epsilon \underset{0}{\underbrace{\frac{\partial z}{\partial n}}}\dot{z}\;ds-\int_{\Omega }H\dot{z}\;dx, \end{array}$$

where we define 
$H:=\epsilon \mathrm{div}\left(\gamma_{*}\nabla z\right)-\frac{\gamma_{*}}{\epsilon }z+\left(1-z\right)W\left(u\right)$
. Since (22b) is written as 
$\alpha \dot{z}={\left(H\right)}_{+}$
, using the equality 
$H{\left(H\right)}_{+}={\left(H\right)}_{+}^{2}$
, we conclude that

$$\begin{array}{cc} & \frac{d}{dt}\left(E_{el}\left(u\left(t\right),z\left(t\right)\right)+E_{s}\left(z\left(t\right)\right)\right)\\ & \;\;\;\;=-\int_{\Omega }H\dot{z}\;dx=-\int_{\Omega }H\frac{{\left(H\right)}_{+}}{\alpha }\;dx=-\int_{\Omega }\frac{{\left(H\right)}_{+}^{2}}{\alpha }\;dx=-\int_{\Omega }\alpha {\left|\dot{z}\right|}^{2}dx. \end{array}$$
 □

### 3.2. Thermal Fracturing Phase Field Model 1 (TF-PFM1)

To combine the Biot model in (1) and (2) and F-PFM in (22), their variational principles for *u*, Proposition 2, and (26) suggest that we consider the following modified thermoelastic energy:
(29)
$$\begin{array}{ccc} E_{el}^{*}\left(u,\Theta ,z\right) & := & \frac{1}{2}\int_{\Omega }{\left(1-z\right)}^{2}\sigma^{*}\left[u,\Theta \right]:e^{*}\left[u,\Theta \right]dx, \end{array}$$

and a variational principle:
(30)
$$u\left(t\right)=\underset{u\in V(u_{D}\left(t\right))}{\mathrm{argmin}}E_{el}^{*}\left(u,\Theta \left(t\right),z\left(t\right)\right).$$


From the definition of the modified thermoelastic energy (29), it is natural to replace the driving force term 
W(u)=σ[u]:e[u]
 in (22b) by the thermoelastic energy density 
$W^{*}\left(u,\Theta \right):=\sigma^{*}\left[u,\Theta \right]:e^{*}\left[u,\Theta \right]$
.

For heat Equation (2), since 
$\beta =a_{L}\left(d\lambda +2\mu \right)$
 and Lamè’s constants (
λ,μ
) are replaced by damaged constants (
${\left(1-z\right)}^{2}\lambda$
, 
${\left(1-z\right)}^{2}\mu$
), 
β
 should also be replaced by damaged constant 
${\left(1-z\right)}^{2}\beta$
. The thermal conductivity 
$\kappa_{0}$
 is also considered to be modified by *z* because the heat is usually insulated across the crack. We suppose 
κ=κ(z)>0
 in this section, and we set it as 
$\kappa \left(z\right)={\left(1-z\right)}^{2}\kappa_{0}$
 in Section 4.

Summarizing the above statements, we obtain the following thermal fracturing model, PFM 1 (TF-PFM1):

$$\left\{ \begin{array}{ccc} -\mathrm{div}\left({\left(1-z\right)}^{2}\sigma^{*}\left[u,\Theta \right]\right)=0 & \mathrm{in}\Omega \times [0,T], & \left(31a\right)\\ \alpha \frac{\partial z}{\partial t}={\left(\epsilon \mathrm{div}\left(\gamma_{*}\nabla z\right)-\frac{\gamma_{*}}{\epsilon }z+\left(1-z\right)W^{*}\left(u,\Theta \right)\right)}_{+} & \mathrm{in}\Omega \times [0,T], & \left(31b\right)\\ \chi \frac{\partial \Theta }{\partial t}=\mathrm{div}\left(\kappa (z)\nabla \Theta \right)-\Theta_{0}{\left(1-z\right)}^{2}\beta \frac{\partial }{\partial t}\left(\mathrm{div}u\right) & \mathrm{in}\Omega \times (0,T], & \left(31c\right) \end{array}\right.$$


Similar to (1), (2) and (22), the boundary and the initial conditions to solve (31) are presented as follows:

$$\left\{ \begin{array}{ccc} u=u_{D}\left(x,t\right) & \mathrm{on}\Gamma_{D}^{u}\times \left[0,T\right], & \left(32a\right)\\ \sigma^{*}\left[u,\Theta \right]n=0 & \mathrm{on}\Gamma_{N}^{u}\times \left[0,T\right], & \left(32b\right)\\ \Theta =\Theta_{D}\left(x,t\right) & \mathrm{on}\Gamma_{D}^{\Theta }\times \left[0,T\right], & \left(32c\right)\\ \frac{\partial \Theta }{\partial n}=0 & \mathrm{on}\Gamma_{N}^{\Theta }\times \left[0,T\right], & \left(32d\right)\\ \frac{\partial z}{\partial n}=0 & \mathrm{on}\Gamma \times [0,T], & \left(32e\right)\\ z\left(x,0\right)=z_{*}\left(x\right) & \mathrm{in}\Omega , & \left(32f\right)\\ \Theta \left(x,0\right)=\Theta_{*}\left(x\right) & \mathrm{in}\Omega . & \left(32g\right) \end{array}\right.$$


In the following, for simplicity, we define

$$\begin{array}{c} \sigma_{z}^{*}\left[u,\Theta \right]:={\left(1-z\right)}^{2}\sigma^{*}\left[u,\Theta \right]. \end{array}$$


As a natural extension of Proposition 2 and Theorem 1, we obtain the following “partial” energy equality for TF-PFM1.

**Theorem** **3**(Energy equality for TF-PFM1)**.**
*We suppose that 
$u_{D}\in H^{\frac{1}{2}}\left(\Gamma_{D}^{u};R^{2}\right)$
 and 
$\Theta \in L^{2}\left(\Theta \right)$
 are given and do not depend on t. If 
u(x,t)
 and 
z(u,t)
 are sufficiency smooth and satisfy (31a), (31b), (32a), (32b), (32e), and (32f), the following energy equality holds:*

(33)
$$\begin{array}{c} \frac{d}{dt}\left(E_{el}^{*}\left(u\left(t\right),\Theta ,z\left(t\right)\right)+E_{s}\left(z\left(t\right)\right)\right)=-\alpha \int_{\Omega }{\left|\dot{z}\right|}^{2}dx\leq 0. \end{array}$$


**Proof.** Under this condition, let us derive 
$E_{el}^{*}\left(u\left(t\right),\Theta ,z\left(t\right)\right)$
 and 
$E_{s}\left(z\left(t\right)\right)$
 with respect to *t*.

(34)
$$\begin{array}{cc} \frac{d}{dt} & \left(E_{el}^{*}\left(u\left(t\right),\Theta ,z\left(t\right)\right)+E_{s}\left(z\left(t\right)\right)\right)\\ & =\frac{1}{2}\frac{d}{dt}\int_{\Omega }\left(\sigma_{z}^{*}\left[u,\Theta \right]:e^{*}\left[u,\Theta \right]\right)dx+\frac{1}{2}\frac{d}{dt}\int_{\Omega }\gamma_{*}\left(\epsilon |\nabla z{|}^{2}+\frac{{\left|z\right|}^{2}}{\epsilon }\right)dx\\ & =\int_{\Omega }\sigma_{z}^{*}\left[u,\Theta \right]:e\left[\dot{u}\right]dx+\int_{\Omega }\left(\gamma_{*}\epsilon \nabla z\cdot \nabla \dot{z}+\left(\frac{\gamma_{*}}{\epsilon }z-\left(1-z\right)W^{*}\left(u,\Theta \right)\right)\dot{z}\right)dx\\ & =\int_{\Gamma }\underset{0}{\underbrace{\sigma_{z}^{*}\left[u,\Theta \right]n}}\cdot e\left[\dot{u}\right]ds-\int_{\Omega }\underset{0}{\underbrace{\mathrm{div}\sigma_{z}^{*}\left[u,\Theta \right]}}\cdot e\left[\dot{u}\right]dx+\gamma_{*}\epsilon \int_{\Gamma }\underset{0}{\underbrace{\frac{\partial z}{\partial n}}}\dot{z}\;ds-\int_{\Omega }H^{*}\dot{z}\;dx, \end{array}$$

where we also define 
$H^{*}:=\epsilon \mathrm{div}\left(\gamma_{*}\nabla z\right)-\frac{\gamma_{*}}{\epsilon }z+\left(1-z\right)W^{*}\left(u,\Theta \right)$
. Since (31b) is changed to 
$\alpha \dot{z}={\left(H^{*}\right)}_{+}$
, similar to that in Section 3.1, we conclude that

$$\begin{array}{cc} & \frac{d}{dt}\left(E_{el}^{*}\left(u\left(t\right),\Theta ,z\left(t\right)\right)+E_{s}\left(z\left(t\right)\right)\right)=-\alpha \int_{\Omega }{\left|\dot{z}\right|}^{2}dx\leq 0, \end{array}$$

which is equivalent to (33). □

### 3.3. Thermal Fracturing Phase Field Model 2 (TF-PFM2)

In the previous section, we proposed TF-PFM1 based on the thermoelastic energy 
$E_{el}^{*}\left(u,\Theta \right)$
. We proved a variational principle but proved only partial energy equality. As shown in Section 2.2, the Biot model is related to both energies 
$E_{el}^{*}\left(u,\Theta \right)$
 and 
$E_{el}\left(u\right)$
. The variational principle holds for 
$E_{el}^{*}\left(u,\Theta \right)$
 (Proposition 2), and the energy equality holds for 
$E_{el}\left(u\right)$
 (Theorem 1). This motivates us to consider another type of thermal fracturing PFM based on elastic energy 
$E_{el}\left(u\right)$
. We call the following thermal fracturing model TF-PFM2:

$$\left\{ \begin{array}{ccc} -\mathrm{div}\left({\left(1-z\right)}^{2}\sigma^{*}\left[u,\Theta \right]\right)=0 & \mathrm{in}\Omega \times [0,T], & \left(35a\right)\\ \alpha \frac{\partial z}{\partial t}={\left(\epsilon \mathrm{div}\left(\gamma_{*}\nabla z\right)-\frac{\gamma_{*}}{\epsilon }z+\left(1-z\right)W\left(u\right)\right)}_{+} & \mathrm{in}\Omega \times [0,T], & \left(35b\right)\\ \chi \frac{\partial \Theta }{\partial t}=\mathrm{div}\left(\kappa (z)\nabla \Theta \right)-\Theta_{0}{\left(1-z\right)}^{2}\beta \frac{\partial }{\partial t}\left(\mathrm{div}u\right) & \mathrm{in}\Omega \times (0,T]. & \left(35c\right) \end{array}\right.$$


The associated boundary and initial conditions are given by (32). For this model, we can show the following energy equality.

**Theorem** **4**(Energy equality for TF-PFM2)**.**
*We suppose that 
(u(x,t),

Θ(x,t),z(x,t))
 is a sufficiently smooth solution for (35) and (32). If 
$u_{D}$
 is independent of t and 
$\Theta_{D}=\Theta_{0}$
, then the following energy equality holds:*

(36)
$$\begin{array}{cc} & \frac{d}{dt}\left(E_{el}\left(u\left(t\right),z\left(t\right)\right)+E_{s}\left(z\left(t\right)\right)+E_{th}\left(\Theta \left(t\right)\right)\right)=-\frac{1}{\Theta_{0}}\int_{\Omega }\kappa \left(z\right){\left|\nabla \Theta \right|}^{2}dx-\alpha \int_{\Omega }{\left|\dot{z}\right|}^{2}dx\leq 0. \end{array}$$


**Proof.** Since the relation in (13) is written as

$$\begin{array}{cc} & \frac{d}{dt}\left(\frac{1}{2}W\left(u\right)\right)=\sigma^{*}\left[u,\Theta \right]:e\left[\dot{u}\right]+\beta \left(\Theta -\Theta_{0}\right)\mathrm{div}\dot{u}, \end{array}$$

we obtain

$$\begin{array}{cc} & \frac{d}{dt}\left(\frac{1}{2}{\left(1-z\right)}^{2}W\left(u\right)\right)=\sigma_{z}^{*}\left[u,\Theta \right]:e\left[\dot{u}\right]+\beta {\left(1-z\right)}^{2}\left(\Theta -\Theta_{0}\right)\mathrm{div}\dot{u}-\left(1-z\right)\dot{z}W\left(u\right). \end{array}$$
Hence, we have

(37)
$$\begin{array}{cc} \frac{d}{dt}E_{el}\left(u\left(t\right),z\left(t\right)\right) & +\frac{d}{dt}E_{s}\left(z\left(t\right)\right)\\ & =\int_{\Omega }\frac{d}{dt}\left(\frac{1}{2}{\left(1-z\right)}^{2}W\left(u\right)\right)dx+\int_{\Omega }\left(\epsilon \mathrm{div}\left(\gamma_{*}\nabla z\right)-\frac{\gamma_{*}}{\epsilon }z\right)\dot{z}\;dx\\ & =\underset{0}{\underbrace{\int_{\Omega }\sigma_{z}^{*}\left[u,\Theta \right]:e\left[\dot{u}\right]dx}}+\int_{\Omega }\beta {\left(1-z\right)}^{2}\left(\Theta -\Theta_{0}\right)\mathrm{div}\dot{u}\;dx-\int_{\Omega }H\dot{z}\;dx\\ & =\int_{\Omega }\beta {\left(1-z\right)}^{2}\left(\Theta -\Theta_{0}\right)\mathrm{div}\dot{u}\;dx-\int_{\Omega }\alpha {\left|\dot{z}\right|}^{2}\;dx, \end{array}$$

where 
$H=\epsilon \mathrm{div}\left(\gamma_{*}\nabla z\right)-\frac{\gamma_{*}}{\epsilon }z+\left(1-z\right)W\left(u\right)$
.On the other hand,

(38)
$$\begin{array}{cc} \frac{d}{dt}E_{th}\left(\Theta \left(t\right)\right) & =\frac{\chi }{\Theta_{0}}\int_{\Omega }\left(\Theta -\Theta_{0}\right)\dot{\Theta }\;dx\\ & =\frac{1}{\Theta_{0}}\int_{\Omega }\left(\Theta -\Theta_{0}\right)\left\{\mathrm{div}\left(\kappa \left(z\right)\nabla \Theta \right)-\Theta_{0}\beta {\left(1-z\right)}^{2}\mathrm{div}\dot{u}\right\}\;dx\\ & =-\frac{1}{\Theta_{0}}\int_{\Omega }\kappa \left(z\right){\left|\nabla \Theta \right|}^{2}\;dx-\int_{\Omega }\beta {\left(1-z\right)}^{2}\left(\Theta -\Theta_{0}\right)\mathrm{div}\dot{u}\;dx. \end{array}$$
Taking a sum of these equalities (37) and (38), then we obtain the energy equality (36). □

## 4. Numerical Experiments

In this section, we conduct numerical experiments to test F-PFM, TF-PFM1, and TF-PFM2, which were derived in Section 3, and report the numerical results. Through the numerical experiments, we observe the effect of thermal coupling on the crack speed and the crack path during its growth process.

### 4.1. Non-Dimensional Setting

In the following numerical examples, we suppose 
$\kappa \left(z\right)={\left(1-z\right)}^{2}\kappa_{0}$
. For convenience, we consider the non-dimensional form with (17), (18), (20) and

$$\tilde{\epsilon }=\frac{\epsilon }{c_{x}},\;{\tilde{\gamma }}_{*}=\frac{c_{e}\gamma_{*}}{c_{x}{\left(\beta c_{\Theta }\right)}^{2}},\;\tilde{\alpha }=\frac{c_{e}\alpha }{c_{t}{\left(\beta c_{\Theta }\right)}^{2}},\;{\tilde{a}}_{L}=\frac{c_{x}c_{\Theta }}{c_{u}}a_{L},\;\tilde{\beta }=1.$$


Then, TF-PFM1 in (31) is expressed in the following non-dimensional form:

$$\left\{ \begin{array}{ccc} \mathrm{div}({\left(1-z\right)}^{2}\sigma \left[u\right])={\left(1-z\right)}^{2}\nabla \Theta & \mathrm{in}\Omega \times [0,T], & \left(39a\right)\\ \alpha \frac{\partial z}{\partial t}={\left(\epsilon \mathrm{div}\left(\gamma_{*}\nabla z\right)-\frac{\gamma_{*}}{\epsilon }z+\left(1-z\right)W^{*}\left(u,\Theta \right)\right)}_{+} & \mathrm{in}\Omega \times [0,T], & \left(39b\right)\\ \frac{\partial \Theta }{\partial t}=\mathrm{div}\left({\left(1-z\right)}^{2}\nabla \Theta \right)-{\left(1-z\right)}^{2}\delta \frac{\partial }{\partial t}\left(\mathrm{div}u\right) & \mathrm{in}\Omega \times (0,T]. & \left(39c\right) \end{array}\right.$$


For TF-PFM2, we change (39b) to:
(40)
$$\alpha \frac{\partial z}{\partial t}={\left(\epsilon \mathrm{div}\left(\gamma_{*}\nabla z\right)-\frac{\gamma_{*}}{\epsilon }z+\left(1-z\right)W\left(u\right)\right)}_{+}\;\;\mathrm{in}\Omega \times [0,T].$$


### 4.2. Time Discretization

To solve problem (39), we adopt the following semi-implicit time discretization scheme [22,26]:

$$\left\{ \begin{array}{cc} -\mathrm{div}\left({\left(1-z^{k-1}\right)}^{2}\sigma^{*}\left[u^{k},\Theta^{k-1}\right]\right)=0, & \left(41a\right)\\ \alpha \frac{{\tilde{z}}^{k}-z^{k-1}}{\Delta t}=\epsilon \mathrm{div}\left(\gamma_{*}\nabla {\tilde{z}}^{k}\right)-\frac{\gamma_{*}}{\epsilon }{\tilde{z}}^{k}+\left(1-{\tilde{z}}^{k}\right)W^{*}\left(u^{k-1},\Theta^{k-1}\right), & \left(41b\right)\\ z^{k}:=\max\left({\tilde{z}}^{k},z^{k-1}\right), & \left(41c\right)\\ \frac{\Theta^{k}-\Theta^{k-1}}{\Delta t}=\mathrm{div}\left(\left(1-z^{k-1}\right)\nabla \Theta^{k}\right)-\left(1-z^{k-1}\right)\delta \mathrm{div}\left(\frac{u^{k}-u^{k-1}}{\Delta t}\right). & \left(41d\right) \end{array}\right.$$


For TF-PFM2, Ref. (41b) is replaced by

(42)
$$\begin{array}{cc} & \alpha \frac{{\tilde{z}}^{k}-z^{k-1}}{\Delta t}=\epsilon \mathrm{div}\left(\gamma_{*}\nabla {\tilde{z}}^{k}\right)-\frac{\gamma_{*}}{\epsilon }{\tilde{z}}^{k}+\left(1-{\tilde{z}}^{k}\right)W\left(u^{k-1}\right), \end{array}$$

where 
$u^{k}$
, 
$z^{k}$
, and 
$\Theta^{k}$
 are the approximations of *u*, *z*, 
Θ
, respectively, at time 
$t_{k}:=k\Delta t\left(k=1,2,3,\ldots \right)$
. Since the adaptive mesh technique in the FEM is often effective and accurate in numerical experiments with phase field models, problems (41) and (42) are calculated using adaptive finite elements with P2 elements with a minimum mesh size of 
$h_{min}=2\times 10^{-3}$
 and a maximum mesh size of 
$h_{max}=0.1$
. The adaptive mesh control at each time step is performed by the adaptmesh() command in FreeFEM based on the variable *z*. An example of the adaptive mesh is illustrated in Figure 9 right. In addition, the code for the following numerical experiments in the current study is written on FreeFEM [39] and executed on a desktop with an Intel(R) Core i7−7820X CPU@3.60 GHz, 16 core processor, and 64 GB RAM.

### 4.3. Thermoelastic Effect on the Crack Speed

We set a square domain 
$\Omega :={\left(-1,1\right)}^{2}\subset R^{2}$
 with the initial crack 
$z_{*}\left(x\right):=e^{\left(-{\left(x_{2}/\eta \right)}^{2}\right)}$

$/(1+e^{\left(x_{1}/\eta \right)})$
 and 
$\eta =1.5\times 10^{-2}$
. The initial mesh is adapted to 
$z_{*}\left(x\right)$
, as illustrated in Figure 9 right. The material constants for the following examples in the non-dimensional form are listed in Table 4.

The boundary conditions for *u* and 
Θ
 are illustrated in Figure 9 left. For *z*, we set 
$\frac{\partial z}{\partial n}=0$
 on 
Γ
.

In Figure 10, the numerical results obtained by F-PFM, TF-PFM1, and TF-PFM2 are shown in the upper, middle, and bottom parts, respectively, where we set 
δ=0.5
 for TF-PFM1 and TF-PFM2. In addition, the profile of *z* on line 
$x_{2}=0$
 is shown in Figure 11. From Figure 10 and Figure 11, we observe that the crack propagation rate obtained by F-PFM is slower than that obtained by the others, and that the crack propagation rate obtained by TF-PFM1 is slightly faster than that obtained by TF-PFM2.

The temperature distributions obtained by TF-PFM1 and TF-PFM2 are shown in Figure 12. In the equation for 
Θ
, the heat resource is given by 
$-{\left(1-z\right)}^{2}\delta \frac{d}{dt}\left(\mathrm{div}u\right)$
. During crack propagation 
(0.4≤t≤0.8)
, the areas near the crack tip, the upper-right corner, and lower-right corner are continuously expanding when 
divu>0
 and 
$\frac{\partial }{\partial t}\left(\mathrm{div}u\right)>0$
. Therefore, due to the negative source 
$-\frac{\partial }{\partial t}\left(\mathrm{div}u\right)$
, lower temperatures are observed in those areas. On the other hand, at 
t=1
, due to the sudden compression caused by the total fracture, positive heat is generated, and a higher temperature is observed, especially near the upper-right and lower-right corners. In this condition, it does not allow for temperature discontinuities along the crack even if we set 
$\kappa \left(z\right)={\left(1-z\right)}^{2}\kappa_{0}$
.

To see how the thermoelastic coupling parameters contribute to enhanced crack propagation, we consider 
δ=0,0.1,0.2,0.5
 for TF-PFM1 and TF-PFM2, and their elastic and surface energies are plotted in Figure 13. From Figure 13, we observe that faster crack propagation occurs with a larger coupling parameter. The figure also shows that crack propagation using TF-PFM1 is faster than that using TF-PFM2.

### 4.4. Thermoelastic Effect on the Crack Path

In this section, we investigate the effect of the thermoelastic coupling parameter on crack path selection using our proposed models. Under a given temperature gradient, we consider crack propagation of an opening mode (Mode I) and a mixed mode (Mode I + II). In the following numerical examples, we also use the parameters in Table 4.

#### 4.4.1. Mode I

We use an edge-cracked square domain, which is shown in Figure 14 left. We set the domain as follows:
$$\begin{array}{cc} & C_{\pm }:=\left(\begin{array}{c} -\frac{1}{2}\\ \pm \frac{5}{8} \end{array}\right)\in R^{2},\\ & H_{\pm }:=\left\{x\in R^{2};\left|x-C_{\pm }\right|\leq \frac{3}{20}\right\},\\ & \Omega :={\left(-1,1\right)}^{2}\backslash \left(H_{+}\cup H_{-}\right), \end{array}$$

and we define

$$\begin{array}{cc} & \Gamma_{DN1}^{u}:=\Gamma \cap \left\{x_{1}=1\right\},\Gamma_{DN2}^{u}:=\partial H_{+}\cup \partial H_{-},\Gamma_{N}^{u}:=\Gamma \backslash \left(\Gamma_{DN1}^{u}\cup \Gamma_{DN2}^{u}\right),\\ & \Gamma_{\pm D}^{\Theta }:=\Gamma \cap \left\{x_{2}=\pm 1\right\},\Gamma_{N}^{\Theta }:=\Gamma \backslash \left(\Gamma_{+D}^{\Theta }\cup \Gamma_{-D}^{\Theta }\right). \end{array}$$


The boundary conditions for *u* and 
Θ
 are given as follows:
$$\begin{array}{c} \left\{\begin{array}{c} u_{1}=0\\ \sigma_{12}^{*}=0 \end{array}\right.\mathrm{on}\Gamma_{DN1}^{u},\;\left\{\begin{array}{c} \left(\sigma^{*}n\right)\cdot e_{1}=0\\ u_{2}=\pm 8t \end{array}\right.\mathrm{on}\partial H_{\pm },\;\sigma^{*}\left[u,\Theta \right]n=0\mathrm{on}\Gamma_{N}^{u},\\ \Theta =\Theta_{D}\mathrm{on}\Gamma_{+D}^{\Theta },\;\Theta =0\mathrm{on}\Gamma_{-D}^{\Theta },\;\frac{\partial \Theta }{\partial n}=0\mathrm{on}\Gamma_{N}^{\Theta }. \end{array}$$


The initial condition for 
Θ
 is given as 
$\Theta_{*}=0$
.

For *z*, similar to the previous example (Section 4.3), we set 
$\frac{\partial z}{\partial n}=0$
 on 
Γ
 and choose the initial value as 
$z_{*}\left(x\right):=e^{\left(-{\left(x_{2}/\eta \right)}^{2}\right)}/\left(1+e^{\left(\left(x_{1}+0.2\right)/\eta \right)}\right)$
 with 
$\eta =1.5\times 10^{-2}$
. In this numerical experiment, we apply the thermoelastic coupling parameter 
δ=0.5
.

Figure 15 shows the different crack paths obtained by the three models when 
$\Theta_{D}=10$
. Straight cracks occur in the F-PFM path since the thermal effect is ignored there. On the other hand, crack curves occur in the TF-PFM1 and TF-PFM2 paths. Here, the crack path is more curved in the TF-PFM2 path than in the TF-PFM1 path. These results show good qualitative agreement with the results reported in [44].

Figure 16 shows the crack paths for different temperature gradients 
$\Theta_{D}=0,3,5,7,10$
 obtained by TF-PFM1 (left) and TF-PFM2 (right). A larger temperature gradient generates a more curved crack path, and TF-PFM2 obtains a more curved crack path than TF-PFM1. Both have significant differences in the magnitude of angle deviation but have the same crack path directions. Therefore, it is clear that thermal expansion changes the crack path.

The temperature distributions during crack growth are shown in Figure 17. There exists a temperature discontinuity along the crack path, which is caused by 
$\kappa \left(z\right)={\left(1-z\right)}^{2}\kappa_{0}$
. It approximately represents a thermal insulation condition across the crack. Different from the previous condition in Section 4.3, although we involve 
$\frac{\partial }{\partial t}\left(\mathrm{div}u\right)$
, its contribution is small.

#### 4.4.2. Mode I + II

According to the numerical experiment in [26], we consider the following setting for mixed mode crack propagation under a thermal gradient. Let 
$\Omega :={\left(-1,1\right)}^{2}\in R^{2}$
, as shown in Figure 14 right, and 
Γ:=∂Ω
. We set

$$\begin{array}{cc} & \Gamma_{\pm D}^{u}:=\Gamma \cap \left\{x_{2}=\pm 1\right\},\Gamma_{N}^{u}:=\Gamma \backslash \left(\Gamma_{+D}^{u}\cup \Gamma_{-D}^{u}\right),\\ & \Gamma_{\pm D}^{\Theta }:=\Gamma \cap \left\{x_{2}=\pm 1\right\},\Gamma_{N}^{\Theta }:=\Gamma \backslash \left(\Gamma_{+D}^{\Theta }\cup \Gamma_{-D}^{\Theta }\right). \end{array}$$


The boundary conditions for *u* are given as follows:
$$\begin{array}{cc} & \left\{\begin{array}{c} u_{1}=\pm 3\sin\left(\pi /3\right)t,\\ u_{2}=\pm 3\cos\left(\pi /3\right)t \end{array}\right.\mathrm{on}\Gamma_{\pm D}^{u},\;\sigma^{*}\left[u,\Theta \right]n=0\mathrm{on}\Gamma_{N}^{u}. \end{array}$$


The boundary conditions for 
Θ
 and *z* are the same as those in Section 4.4.1. The initial crack profile is given as 
$z_{*}\left(x\right):=e^{\left(-{\left(x_{2}/\eta \right)}^{2}\right)}/\left(1+e^{\left(\left(x_{1}-0.5\right)/\eta \right)}\right)-e^{\left(-{\left(x_{2}/\eta \right)}^{2}\right)}/\left(1+e^{\left(\left(x_{1}+0.5\right)/\eta \right)}\right)$
 with 
$\eta =1.5\times 10^{-2}$
. We fix the thermoelastic coupling parameter 
δ=0.15
 and change the temperature gradient to 
$\Theta_{D}=0,2,3,5,6$
.

Figure 18 shows the crack paths obtained by TF-PFM1 and TF-PFM2. The cracks are kinked, and the kink angle becomes larger when the thermal gradient 
$\Theta_{D}$
 increases. The two models provide similar results, but the kink angle in the TF-PFM2 crack is larger than that in the TF-PFM1 crack, as shown in Figure 19. Therefore, we conclude that thermal expansion changes the crack path.

Here, we do not show the temperature distribution during thermal expansion. We observe that the temperature distribution is quite similar to that of Mode I in Section 4.4.1, and a temperature discontinuity exists along the crack path during temperature injection. As mentioned, since it is relatively difficult to find the available experimental result for the thermal fracturing under mode I + II, we do not compare our result with the experimental result.

At the end of this section, we give a remark on the extendability of our TF-PFM to anisotropic material. When the material has a strong anisotropy, we have to take into account the anisotropies on the elasticity tensor *C*, a coefficient of linear thermal expansion 
$a_{L}$
, and the critical energy release rate 
$\gamma_{*}$
, especially among many material properties. For *C* and 
$a_{L}$
, we can easily include an anisotropic effect by using an anisotropic tensor in (3b) and replace the matrix 
$a_{L}I$
 in (4a) by anisotropic one. On the other hand, for 
$\gamma_{*}$
, it has not well succeeded to include the anisotropy, which means that the dependency of the crack surface direction, even in the standard PFM, as far as the authors’ knowledge.

## 5. Conclusions and Future Works

We proposed two thermal fracturing phase field models, TF-PFM1 and TF-PFM2, by coupling the Biot thermoelasticity model [8] and the fracturing phase field model (F-PFM) by Takaishi–Kimura [22,26].

For the Biot model, we studied a variational principle (Proposition 2) and energy equality (Theorem 1), which were related to different energies 
$E_{el}^{*}\left(u,\Theta \right)$
 and 
$E_{el}\left(u\right)+E_{th}\left(\Theta \right)$
, respectively (see Table 1 and Table 2).

On the other hand, F-PFM has a gradient flow structure with respect to the total energy 
$E_{el}\left(u,z\right)+E_{s}\left(z\right)$
 and admits energy equality (Theorem 2).

As the first model, TF-PFM1 was derived based on the variational principle of the Biot model and the gradient flow structure of F-PFM, while TF-PFM2 is based on the energy equalities of the Biot model and F-PFM. The difference between them is the driving force term for the crack: 
$W^{*}\left(u,\Theta \right)$
 in TF-PFM1 (31b) and 
W(u)
 in TF-PFM2 (35b).

Consequently, we established partial energy equality for TF-PFM1 (Theorem 3) and energy equality for TF-PFM2 (Theorem 4). From the viewpoint of energy consistency, both models are satisfactory, but TF-PFM2 is more energetically consistent than TF-PFM1.

Based on the obtained numerical experiments, the following conclusions can be drawn.

The thermoelastic coupling parameter 
δ
 in TF-PFM1 and TF-PFM2 enhances crack propagation (Figure 10).TF-PFM1 accelerates the crack speed more than TF-PFM2 (Figure 11). On the other hand, the effect of the temperature gradient on the crack path in TF-PFM2 is larger than that in TF-PFM1 (Figure 16, Figure 17, Figure 18 and Figure 19).

The analytical and numerical comparisons between the two models are briefly summarized in Table 5.

In this study, we did not consider the unilateral contact condition along the crack for the sake of simplicity. To further improve TF-PFM, the ideal unilateral condition for fracturing PFM [21,26] should be introduced in our PFM. In addition, there are many other effects that should be included in the model. For example, although we assumed that the critical energy release rate 
$\gamma_{*}\left(x\right)$
 is a priory given, it may depend on the temperature in the real material. A possible extension of our model is to suppose that 
$\gamma_{*}$
 depends on 
Θ
 linearly as

$$\begin{array}{c} \gamma_{*}\left(x,\Theta \right)=\bar{\gamma }\left(1-\alpha_{0}\left(\Theta -\Theta_{0}\right)\right) \end{array}$$

for some 
$\bar{\gamma }>0$
 and 
α>0
 [45].

Such relatively easy extendability is one of the advantages of PFM. However, we should remark that the energy equalities which we derived in this paper may not be valid for all extended models.

## Figures and Tables

**Figure 1 materials-15-02571-f001:**
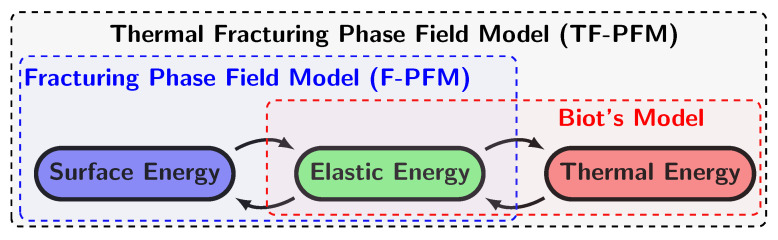
A conceptual diagram of energy balance for Biot’s model, F-PFM, and TF-PFM.

**Figure 2 materials-15-02571-f002:**
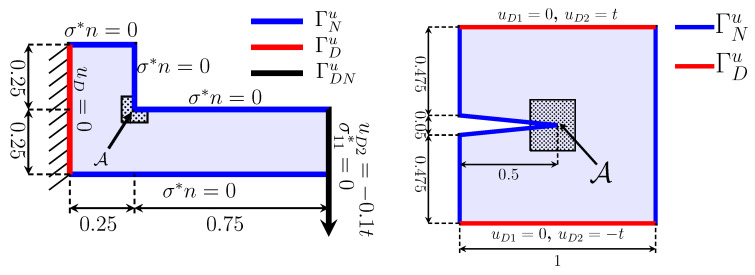
An L-shaped cantilever (**left**) and a cracked domain (**right**) with the subdomain 
A
 as an observation area.

**Figure 3 materials-15-02571-f003:**
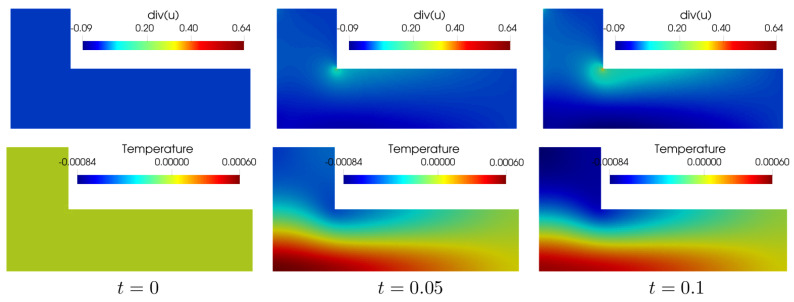
Snapshots of 
divu
 (**upper**) and the temperature (**lower**) of the L-shape cantilever for 
t=0,0.05,0.1
 using 
δ=0.1
. Near the re-entrant corner, the domain is expanded (
divu>0
), and the temperature decreases. On the other hand, near the bottom boundary, the domain is compressed (
divu<0
), and the temperature increases.

**Figure 4 materials-15-02571-f004:**
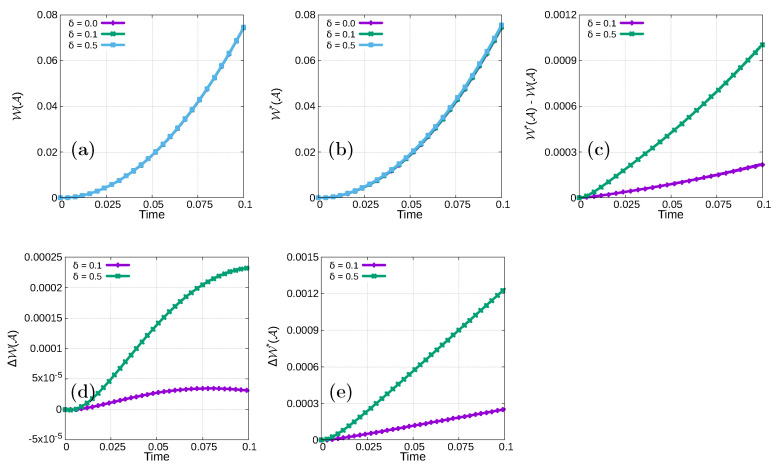
Profile of (**a**) 
W(A)
, (**b**) 
$W^{*}\left(A\right)$
, (**c**) 
$W^{*}\left(A\right)-W\left(A\right)$
, (**d**) 
ΔW(A)
 and (**e**) 
$\Delta W^{*}\left(A\right)$
 in an L-shaped cantilever during the loading process.

**Figure 5 materials-15-02571-f005:**
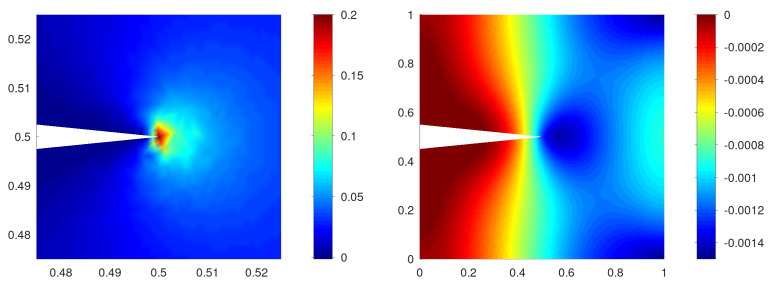
Snapshots of 
divu
 on the subdomain 
A
 (**left**) and temperature 
Θ
 in 
Ω
 (**right**) using 
δ=0.1
 at 
t=0.1
.

**Figure 6 materials-15-02571-f006:**
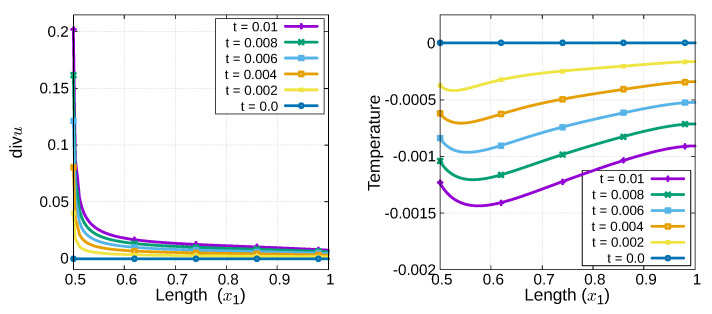
Profile of 
divu
 (**left**) and temperature 
Θ
 (**right**) using 
δ=0.1
 along the 
$x_{1}$
 axis, i.e., 
$x_{2}=0$
, 
$0.5\leq x_{1}\leq 1$
, during the loading process.

**Figure 7 materials-15-02571-f007:**
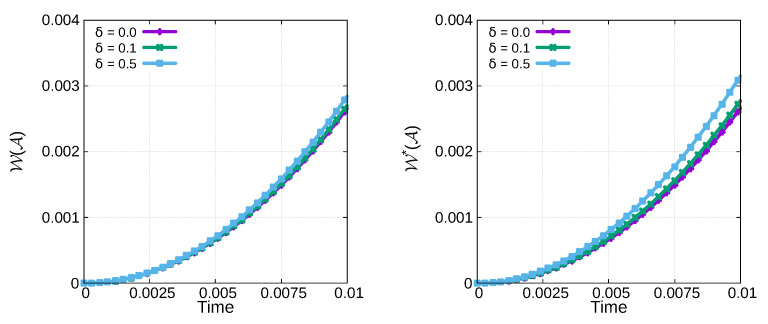
Profile of 
W(A)
 (**left**) and 
$W^{*}\left(A\right)$
 (**right**) in subdomain 
A
 during the loading process.

**Figure 8 materials-15-02571-f008:**
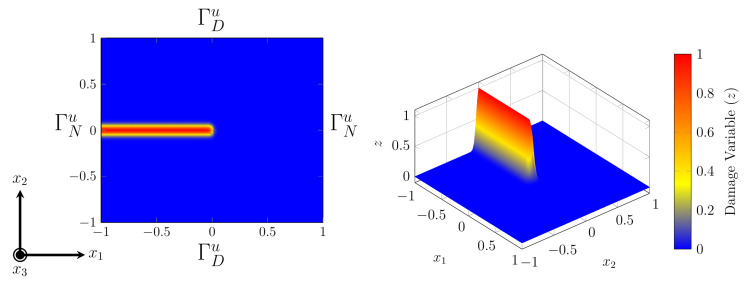
Illustration of the phase field approximation of the cracked surface in an elastic body.

**Figure 9 materials-15-02571-f009:**
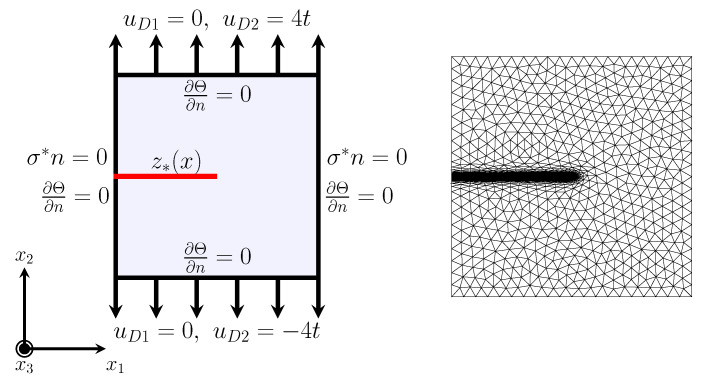
Domain for Section 4.3 with 
$z_{*}\left(x\right)$
 as the initial crack (**left**) and the adaptive mesh for the initial crack (**right**).

**Figure 10 materials-15-02571-f010:**
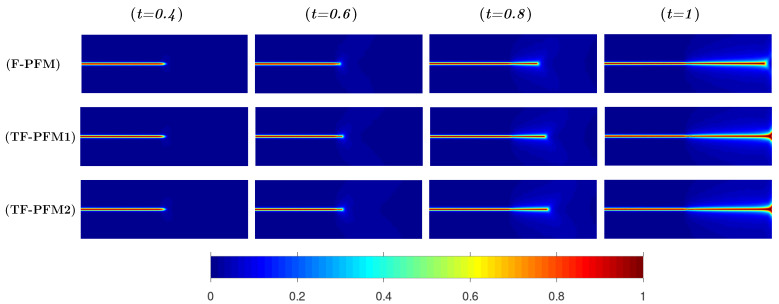
Snapshots of crack propagation with F-PFM, TF-PFM1, and TF-PFM2 in 
(−1,1)×(−0.35,0.35)
 at 
t=0.4,0.6,0.8,1
 (left to right). For TF-PFM1 and TF-PFM2, we use the thermoelasticity coupling parameter 
δ=0.5
, and the color represents the value of *z*.

**Figure 11 materials-15-02571-f011:**
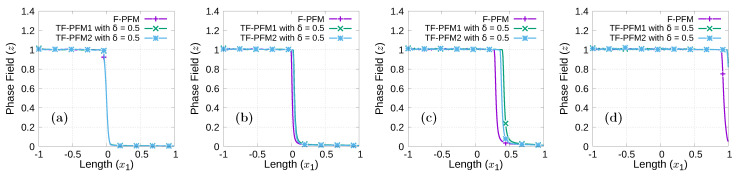
Comparison of the profiles of *z* obtained by F-PFM, TF-PFM1, and TF-PFM2 along the line 
$x_{2}=0$
 at (**a**) 
t=0.4
, (**b**) 
t=0.6
, (**c**) 
t=0.8
, and (**d**) 
t=1
.

**Figure 12 materials-15-02571-f012:**
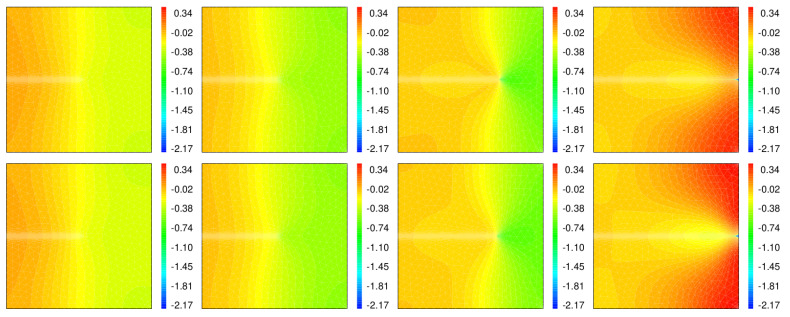
Snapshots of the temperatures obtained by TF-PFM1 (**upper**) and TF-PFM2 (**lower**) at 
t=0.4,0.6,0.8,1
 (**left** to **right**); the color represents the value of 
Θ
.

**Figure 13 materials-15-02571-f013:**
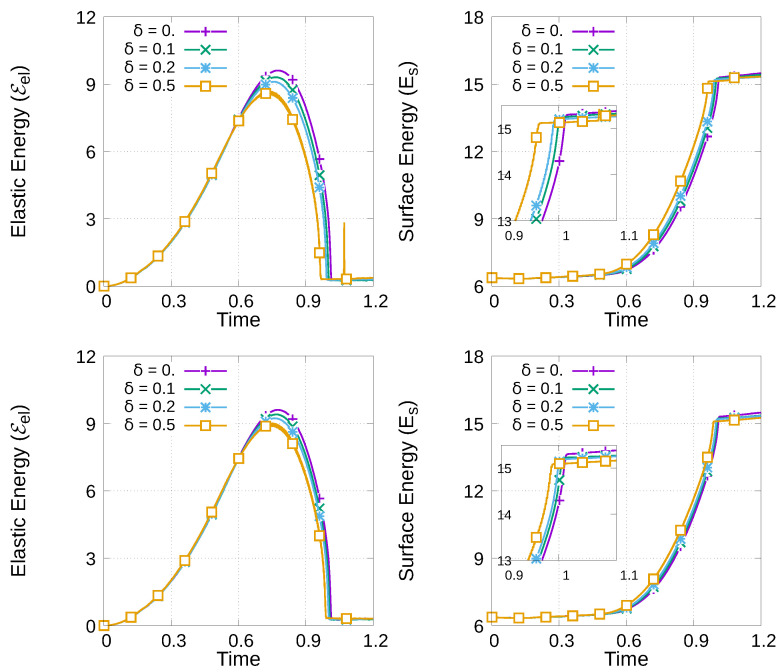
Profile of the elastic (**left**) and surface energy (**right**) under thermal expansion during crack propagation using TF-PFM1 (**top**) and TF-PFM2 (**bottom**).

**Figure 14 materials-15-02571-f014:**
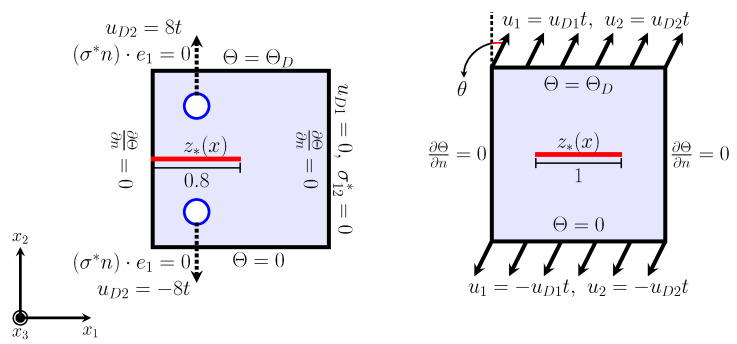
Mode I (**left**) and Mode I + II (**right**) for the study of the crack path under thermal expansion and the loading process. Here, the initial damage 
$z_{*}\left(x\right)$
 is illustrated by the red initial crack in the figures.

**Figure 15 materials-15-02571-f015:**
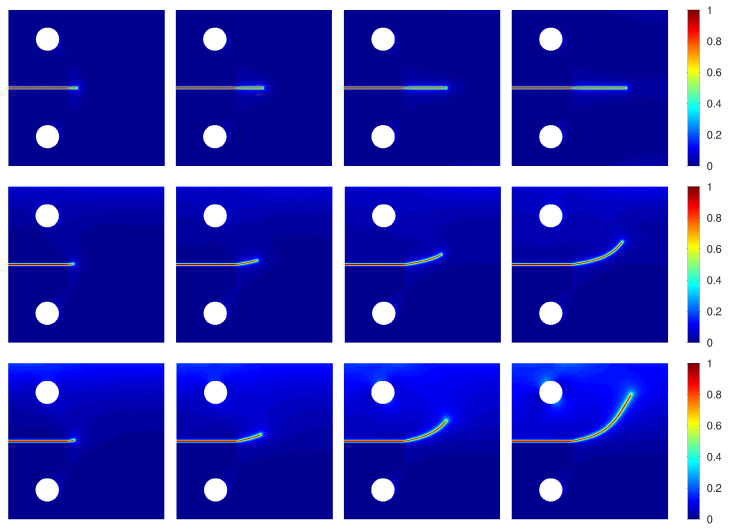
Snapshots of the crack paths. F-PFM (**upper**), TF-PFM1 (**middle**), and TF-PFM2 (**lower**) at 
t=0.4,0.6,0.8,1
 (**left** to **right**). For TF-PFM1 and TF-PFM2, we set 
$\Theta_{D}=10$
 and 
δ=0.5
. Here, the color represents the value of *z*.

**Figure 16 materials-15-02571-f016:**
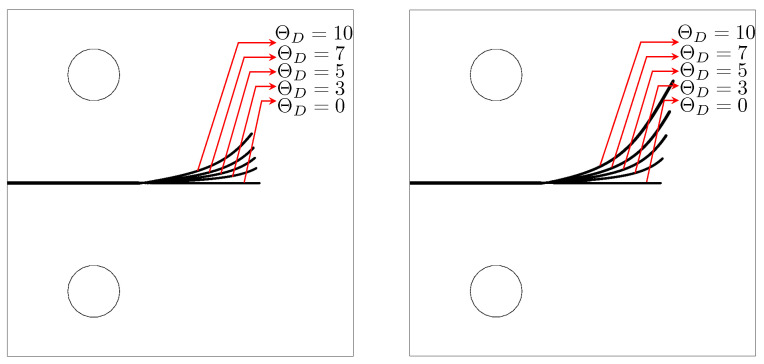
Comparison of the crack paths using TF-PFM1 (**left**) and TF-PFM2 (**right**) with the given temperature variations under Mode I at the final computational time 
t=1
.

**Figure 17 materials-15-02571-f017:**
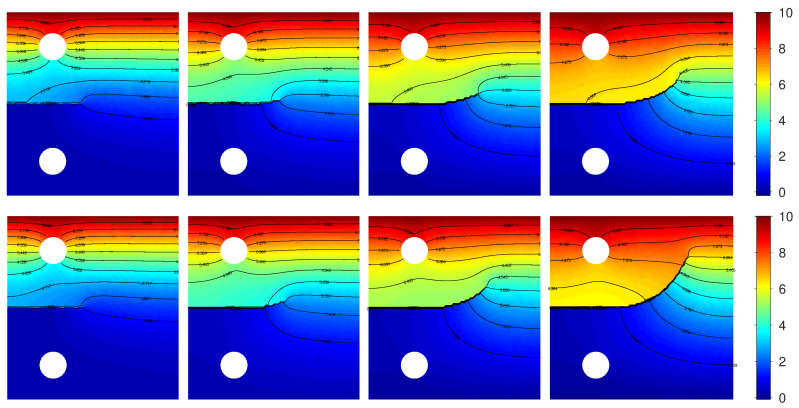
Snapshots of the temperature gradient during thermal expansion and crack growth under the given temperature 
$\Theta_{D}=10$
. TF-PFM1 (**top**) and TF-PFM2 (**bottom**) at 
t=0.4,0.6,0.8,1
 (**left** to **right**); the color represents the value of 
Θ
.

**Figure 18 materials-15-02571-f018:**
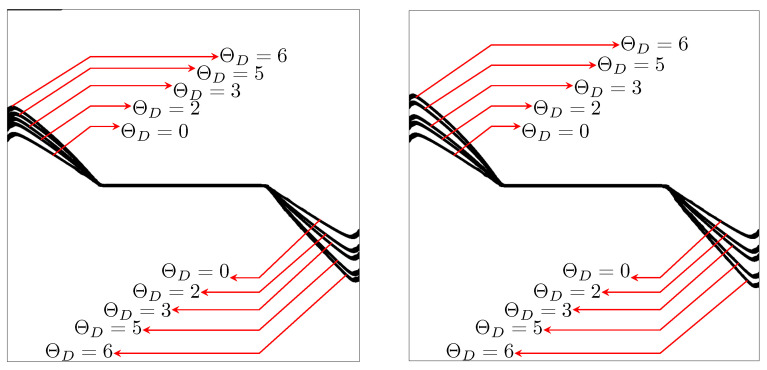
Comparison of the crack paths using TF-PFM1 (**left**) and TF-PFM2 (**right**) with the given temperature variations under Mode I + II at the final computational time.

**Figure 19 materials-15-02571-f019:**
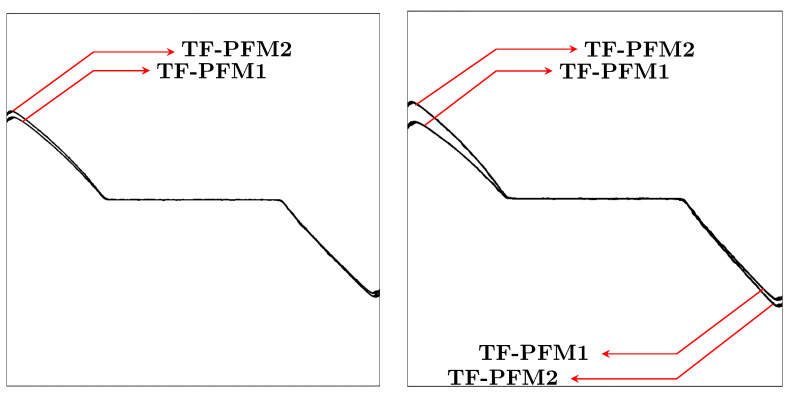
Comparison of the crack paths using TF-PFM1 and TF-PFM2 when 
Θ=5
 (**left**) and 
Θ=6
 (**right**) at the final computational time.

**Table 1 materials-15-02571-t001:** List of energies.

Type of Energy	Definition	Equation
Elastic	$E_{el}\left(u\right):=\frac{1}{2}\int_{\Omega }\sigma \left[u\right]:e\left[u\right]dx$	(8)
Thermoelastic	$E_{el}^{*}\left(u,\Theta \right):=\frac{1}{2}\int_{\Omega }\sigma^{*}\left[u,\Theta \right]:e^{*}\left[u,\Theta \right]dx$	(10)
Thermal	$E_{th}\left(\Theta \right):=\frac{\chi }{2\Theta_{0}}\int_{\Omega }|\Theta \left(x\right)-\Theta_{0}{|}^{2}dx$	(11)
Modified elastic	$E_{el}\left(u,z\right):=\frac{1}{2}\int_{\Omega }{\left(1-z\right)}^{2}\sigma \left[u\right]:e\left[u\right]dx$	(24)
Modified thermoelastic	$E_{el}^{*}\left(u,\Theta ,z\right):=\frac{1}{2}\int_{\Omega }{\left(1-z\right)}^{2}\sigma^{*}\left[u,\Theta \right]:e^{*}\left[u,\Theta \right]dx$	(29)
Surface	$E_{s}\left(z\right):=\frac{1}{2}\int_{\Omega }\gamma_{*}\left({\epsilon |\nabla z|}^{2}+\frac{{\left|z\right|}^{2}}{\epsilon }\right)dx$	(25)

**Table 2 materials-15-02571-t002:** Different forms of energy equalities.

Model	Strong Form	Energy	Energy Equality
Linear elasticity	(6)	$E_{el}\left(u\right)$	-
Biot’s model	(1) and (2)	$E_{el}\left(u\right)+E_{th}\left(\Theta \right)$	(12)
F-PFM	(22a) and (22b)	$E_{el}\left(u,z\right)+E_{s}\left(z\right)$	(27)
TF-PFM1	(31a) and (31c)	$E_{el}^{*}\left(u,\Theta ,z\right)+E_{s}\left(z\right)$	(33) ^a^
TF-PFM2	(35a) and (35c)	$E_{el}\left(u,z\right)+E_{s}\left(z\right)+E_{th}\left(\Theta \right)$	(36)

^a^ When a temperature 
$\Theta =\Theta \left(x\right)\in L^{2}\left(\Omega \right)$
 is given.

**Table 3 materials-15-02571-t003:** List of physical properties.

Symbol	Physical Meaning [Unit]	Symbol	Physical Meaning [Unit]
*u*	Displacement [ m ]	$\sigma^{*}\left[u,\Theta \right]$	Stress tensor with thermal effect [ Pa ]
Θ	Temperature [ K ]	$e^{*}\left[u,\Theta \right]$	Strain tensor with thermal effect [-]
$\Theta_{0}$	Reference temperature [ K ]	β	Stress thermal modulus [ Pa · K ^−1^]
*z*	Damage variable [-]	$\kappa_{0}$	Thermal conductivity [ W · m ^−1^ · K ^−1^]
σ[u]	Stress tensor [ Pa ]	χ	Volumetric heat capacity [ J · K ^−1^ · m ^−3^]
e[u]	Strain tensor [-]	$a_{L}$	Coefficient of linear thermal expansion [ K ^−1^]
$E_{Y}$	Young’s modulus [ Pa ]	δ	Thermoelastic coupling parameter [-]
$\nu_{P}$	Poisson ratio [-]	$\gamma_{*}$	Critical energy release rate ^a^ [ Pa·m ]
λ , μ	Lamé’s constants ^b^ [ Pa ]	ϵ	Length scale in F-PFM or TF-PFM [ m ]
*t*	Time [ s ]	α	Time regularization parameter in F-PFM or TF-PFM [ Pa·s ]

^a^

$\gamma_{*}$
 is usually denoted by 
$G_{c}$
 [26,36]. ^b^

λ
 and 
μ
 are written as 
$\lambda =\frac{E_{Y}\nu_{P}}{\left(1+\nu_{P}\right)\left(1-2\nu_{P}\right)}$
 and 
$\mu =\frac{E_{Y}}{2(1-\nu_{P})}$
.

**Table 4 materials-15-02571-t004:** List of the non-dimensional parameters for Section 4.3 and Section 4.4.

**Parameter**	$E_{Y}$	$\nu_{P}$	$\kappa_{0}$	$a_{L}$	α	ϵ	$\gamma_{*}$	$\Theta_{*}$
**Value**	1	0.3	1.	0.7	0.001	0.01	5.08	0

**Table 5 materials-15-02571-t005:** Numerical comparison of TF-PFM1 and TF-PFM2.

Models	Driving Force	Energy Consistency	Straight Crack Speed	Crack Path
TF-PFM1	$W^{*}\left(u,\Theta \right)=\sigma^{*}\left[u,\Theta \right]:e^{*}\left[u,\Theta \right]$	Partially satisfied	Faster	Less curved
TF-PFM2	W(u)=σ[u]:e[u]	Fully satisfied	Slower	More curved
Remarks	$W^{*}\left(u,\Theta \right)>W\left(u\right)$ (Figure 4)	Theorems 3 & 4	Figure 13	Figure 16 and Figure 19

## Data Availability

Not applicable.

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
