# Peer review of "Phase Field Models for Thermal Fracturing and Their Variational Structures"

_materials, 2022, doi:10.3390/ma15072571_

Round 1
Reviewer 1 Report
The authors performed the phase field models (PFM) for thermal fracturing by coupling the Biot thermoelasticity model and PFM. Before a possible publication, some suggestions that might be helpful for the improvement are listed as follows:
a) The first cited reference in the introduction part is [30], which is very strange. Normally, all the references should be placed in sequence.
b) In the introduction part, the authors have provide some information about the background. However, they should clearly point out what is the moviation of their work? Why should the the Biot thermoelasticity model and PFM combined? If not, are there any problems?
c) Generally speaking, the thermal residual stress could be dependent on the materials properties, for example, which show the different sysmmetries (e.g., cubic and hcp structures). So how those properties affect the thermal fracturing behavior? Could please the authors provide more information about this?
d) For Fig. 5, the crack paths, are there any experimental results from the authors or previous literatures? If yes, the simulation results may be compared with those experimental ones.
e) Finally, after this manuscript well written and organized, it may be more suitable to be submitted to journals like the Mechanics and Physics of Solids for better readship, since the authors have cited many references from those journals and this manuscript involves totally the theoretical framework.
Author Response
Response to Reviewer 1 Comments
Point 1: The first cited reference in the introduction part is [30], which is very strange. Normally, all the references should be placed in sequence.
Response 1: We have changed it sequentially.
Point 2: In the introduction part, the authors have provide some information about the background. However, they should clearly point out what is the motivation of their work? Why should the the Biot thermoelasticity model and PFM combined? If not, are there any problems?
Response 2: We have given a clear motivation for our paper in lines 42 - 43 and an explanation about the combination Biot model dan PFM in lines 50 – 51.
Point 3: Generally speaking, the thermal residual stress could be dependent on the materials properties, for example, which show the different sysmmetries (e.g., cubic and hcp structures). So how those properties affect the thermal fracturing behavior? Could please the authors provide more information about this?
Response 3: Considering these comments, we have put some comments and responses in lines 79 – 82 and 332 – 330.
Point 4: For Fig. 5, the crack paths, are there any experimental results from the authors or previous literatures? If yes, the simulation results may be compared with those experimental ones.
Response 4: Considering this comment, we have explained it in lines 184 – 189.
Point 5: Finally, after this manuscript well written and organized, it may be more suitable to be submitted to journals like the Mechanics and Physics of Solids for better readship, since the authors have cited many references from those journals and this manuscript involves totally the theoretical framework.
Response 5: Refers to your suggestion, we will send the extension of the present study (thermal fracturing in viscous-elastic material) to the Journal of the Mechanics and Physics of Solids.

Reviewer 2 Report
The manuscript presents numerical modelling method for thermal fracture using phase field models. In the opinion of this reviewer, the manuscript is recommended for publication after addressing the followings significant comments.
- As we known, the compression and tension making a different contribution to the phase filed damage evolution. Hence, the inherent elastic energy density is usually decomposed into two parts: the contribution part and non-contribution part. Some decomposition methods have been proposed to deal with different problems. Authors didn’t consider the decomposition of elastic energy density in proposed TF-PFMs.
- In Section 4.4.2, authors didn’t show snapshots of the crack paths and temperature gradient under Mode I+II. In addition, it would be more reasonable if authors had compared the numerical results with the available experimental results.
- In the TF-PFMs proposed by authors, the thermal conductivity k0 is considered, but the defined value of k0 could not be found in section 4.3. Actually, the thermal conductivity affects the temperature distribution of material. It is worth discussing.
- Page 16:please confirm that the equation following the phrase “Proof. Since the relation in (12) is written as” is correct.
- The boundary condition about u2 in Figure 2 is inconsistent with those described in Section 2.3.3.
- page 18; Line 272: “…that obtained by TF-PFM” should be modified to “…that obtained by TF-PFM2”.
- It is worth referring to the recent similar publication:
- Tribology International, 150 (2020) 106384. https://doi.org/10.1016/j.triboint.2020.106384
- Engineering Fracture Mechanics 205 (2019):387-398. https://doi.org/10.1016/j.engfracmech.2018.09.019
Author Response
Response to Reviewer 2 Comments
Point 1: As we known, the compression and tension making a different contribution to the phase filed damage evolution. Hence, the inherent elastic energy density is usually decomposed into two parts: the contribution part and non-contribution part. Some decomposition methods have been proposed to deal with different problems. Authors didn’t consider the decomposition of elastic energy density in proposed TF-PFMs.
Response 1: Our paper does not consider the decomposition of elastic energy density but we put it in line 363 in the “Summary and Future Works” section as future work. As a remark, we use the unilateral contact condition term to represent the decomposition method.
Point 2: In Section 4.4.2, authors didn’t show snapshots of the crack paths and temperature gradient under Mode I+II. In addition, it would be more reasonable if authors had compared the numerical results with the available experimental results.
Response 2: We do not show a snapshot of the crack path but we show the crack path at the final time in Figure 18. We think that it can explain the crack path behavior under thermal expansion. For temperature gradient, we have explained it in lines 3276– 329. For the available experiment results, we have put some comments in lines 329 – 331.
Point 3: In the TF-PFMs proposed by authors, the thermal conductivity k0 is considered, but the defined value of k0 could not be found in section 4.3. Actually, the thermal conductivity affects the temperature distribution of material. It is worth discussing.
Response 3: We have put the value of k0 in Table 4. For the thermal conductivity effects, we have added some explanations in the lines 284 – 285 and 311 – 315.
Point 4: Page 16:please confirm that the equation following the phrase “Proof. Since the relation in (12) is written as” is correct.
Response 4: We have corrected it.
Point 5: The boundary condition about u2 in Figure 2 is inconsistent with those described in Section 2.3.3.
Response 5: We have corrected Figure 2(left).
Point 6: page 18; Line 272: “…that obtained by TF-PFM” should be modified to “…that obtained by TF-PFM2”.
Response 6: We have corrected it. You can see in the line 276
Pont 7: It is worth referring to the recent similar publication:
- Tribology International, 150 (2020) 106384. https://doi.org/10.1016/j.triboint.2020.106384
- Engineering Fracture Mechanics 205 (2019):387-398.
https://doi.org/10.1016/j.engfracmech.2018.09.019
Response 7: We have put them as references for our publication.

Round 2
Reviewer 2 Report
The manuscript is recommended for publication.